# STAR: Similarity-guided Teacher-Assisted Refinement for Super-Tiny Function Calling Models

**Jiliang Ni**[*], **Jiachen Pu**[*], **Zhongyi Yang**[*], **Jingfeng Luo, Conggang Hu**[†]
Qwen Applications Business Group of Alibaba
`conggang.hcg@alibaba-inc.com`

## Abstract

The proliferation of Large Language Models (LLMs) in function calling is pivotal for creating advanced AI agents, yet their large scale hinders widespread adoption, necessitating transferring their capabilities into smaller ones. However, existing paradigms are often plagued by overfitting, training instability, ineffective binary rewards for multi-solution tasks, and the difficulty of synergizing techniques. We introduce STAR: Similarity-guided Teacher-Assisted Refinement, a novel holistic framework that effectively transfers LLMs' capabilities to super-tiny models. STAR consists of two core technical innovations: (1) Constrained Knowledge Distillation (CKD), a training objective that augments top-k forward KL divergence to suppress confidently incorrect predictions, ensuring training stability while preserving exploration capacity for downstream RL. STAR holistically synergizes these strategies within a cohesive training curriculum, enabling super-tiny models to achieve exceptional performance on complex function calling tasks; (2) Similarity-guided RL (Sim-RL), a RL mechanism that introduces a fine-grained, similarity-based reward. This provides a robust, continuous, and rich signal for better policy optimization by evaluating the similarity between generated outputs and the ground truth. Extensive experiments on challenging and renowned benchmarks demonstrate the effectiveness of our method. Our STAR models establish SOTA in their size classes, significantly outperforming baselines. Remarkably, our 0.6B STAR model achieves the best performance among all open models under 1B, surpassing even several well-known open models at a larger scale. STAR demonstrates a training framework that distills capabilities of LLMs into super-tiny models, paving the way for powerful, accessible, and efficient AI agents.

## 1 Introduction

Large Language Models (LLMs) have demonstrated remarkable capabilities as agents that interact with external tools and APIs via function calling (Patil et al., 2024; Jin et al., 2025). This has driven a new generation of applications, from automated personal assistants to complex data analysis systems. However, the prohibitive computational cost of state-of-the-art models driving these advancements, often with tens to hundreds of billions of parameters, hinders their accessibility and practicality for on-device deployment and large-scale services (Guo et al., 2025). This necessitates transferring the capabilities of large models to smaller, more efficient models. However, the conventional strategy (DeepSeek-AI et al., 2025; Cui et al., 2025a) to achieve this, which involves a sequence of Supervised Fine-Tuning (SFT) and Reinforcement Learning (RL), proves inadequate for such super-tiny models. The inherently limited capacity of these models makes them prone to overfitting when trained with SFT on finite, high-quality datasets; they memorize specific tool-use patterns rather than generalize. Concurrently, applying RL directly to small models is notoriously unstable and inefficient (Sarangi & Salam, 2025; Dang & Ngo, 2025).

---

[*]Equal Contribution.
[†]Corresponding Author.

These limitations suggest a more promising approach: combining Knowledge Distillation (KD) to provide a robust, generalizable initialization for RL without the risk of overfitting. Yet, this KD+RL paradigm introduces its own distinct and formidable challenges: (1) **KD instability and constrained exploration:** To manage computational costs, standard KD often employs top-k truncation, leaving the student's long-tail probability distribution unsupervised. This lack of guidance frequently leads to training instability and model collapse, while simultaneously stifling the exploratory capacity essential for the subsequent RL phase; **(2) Ineffective RL rewards:** For multi-solution problems such as function calling, standard discrete or binary success/failure rewards can excessively penalize valid, alternative solutions, thereby impeding effective learning (Wei et al., 2025); (3) **Synergistic integration challenges:** Achieving true synergy between KD and RL, rather than interference, presents a significant practical hurdle.

This context motivates our work, aiming to create an effective and stable training framework that overcomes these obstacles. We introduce STAR: Similarity-guided Teacher-Assisted Refinement, a holistic framework designed to meticulously transfer and refine LLMs' capabilities into super-tiny models. Our contributions are threefold:

- We introduce **Constrained Knowledge Distillation (CKD)**, a novel training objective that enhances top-k forward KL-divergence with a targeted regularization term on the student's probability distribution. This suppresses high-confidence but erroneous predictions without forcing the long-tail distribution to zero, ensuring stability under top-k truncation while preserving the crucial exploratory capacity for downstream RL.

- We propose a novel RL mechanism, **Sim-RL**, that augments the standard task reward with a fine-grained similarity-based reward. This reward is computed from the similarity between generated outputs and the ground truth, providing a robust, continuous, and rich signal to enhance policy optimization without increasing system complexity.

- We present a unified training curriculum that effectively synergizes the strengths of CKD and Sim-RL, culminating in STAR models that establish new SOTA on the challenging and renowned benchmarks for their own sizes. Notably, our 0.6B STAR model achieves relative gains of 9.2% on BFCL and over 50% on ACEBench against baselines. It outperforms all open-source models under 1B and even several significantly larger models. The code link is `https://github.com/Qwen-Applications/STAR`.

The immense inference cost of highly capable large models mainly hinders their large-scale application, making it a critical research goal to elevate small models' performance to near-large-model levels. Our work validates that a well-designed training framework can transfer LLMs' capabilities into super-tiny models. This unlocks their potential in specialized fields, broadens the real-world deployment of advanced AI, and enables the creation of powerful, accessible, and efficient agents.

## 2 TASK DEFINITION: FUNCTION CALLING AS A GENERATION PROBLEM

We formalize the task of function calling as a conditional sequence generation problem. The model is provided with a context, which includes the user's query, a set of available functions $\mathcal{F} = \{f_1, f_2, ..., f_N\}$ and other information. Each function $f_i$ is defined by its name, a description of its purpose, and its parameters.

The model's goal is to generate a sequence of function calls $P = (p_1, p_2, ..., p_n)$ that solves the user's query. A function call is a structured output, typically in a specific format like JSON, e.g., $\{"name" : "...", "arguments" : \{"arg" : "...", ...\}\}$. Additionally, the model is also required to provide natural language responses when no function calls are needed.

## 3 THE STAR METHODOLOGY

The STAR methodology is a comprehensive training framework designed to imbue a super-tiny student model ($M_S$) with the advanced function calling capabilities of a much larger teacher model ($M_T$). It consists of two core technical components—CKD and Sim-RL—applied within a carefully structured training curriculum, as illustrated in Figure 1.

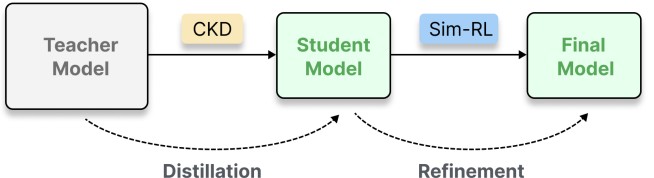

Figure 1: The overview of the STAR training curriculum.

## 3.1 CONSTRAINED KNOWLEDGE DISTILLATION (CKD)

Knowledge distillation (KD) is a cornerstone for aligning a student model ($\mathcal{M}_S$) with a teacher ($\mathcal{M}_T$). A central design choice in KD for language models is the divergence metric, typically oscillating between the distribution-covering Forward KL-divergence ($\mathcal{L}_{\text{FKL}}$) and the mode-seeking Reverse KL-divergence ($\mathcal{L}_{\text{RKL}}$) (Gu et al., 2024; Li et al., 2024). $\mathcal{L}_{\text{RKL}}$ forces the student model ($\mathcal{M}_S$) to focus on the high-probability tokens of the teacher ($\mathcal{M}_T$) while ignoring the vast, often uninformative tail of the distribution, defined as:

$$\mathcal{L}_{\text{FKL}} = \sum_{x \in \mathcal{D}} D_{\text{KL}}(P_T(y|x) \| P_S(y|x)) \tag{1}$$

$$\mathcal{L}_{\text{RKL}} = \sum_{x \in \mathcal{D}} D_{\text{KL}}(P_S(y|x) \| P_T(y|x)) \tag{2}$$

where $P_S$ and $P_T$ represent the output distributions over a vocabulary for a given context $x$. Some methods, like Adaptive Kullback-Leibler divergence (AKL), combine both (Wu et al., 2025).

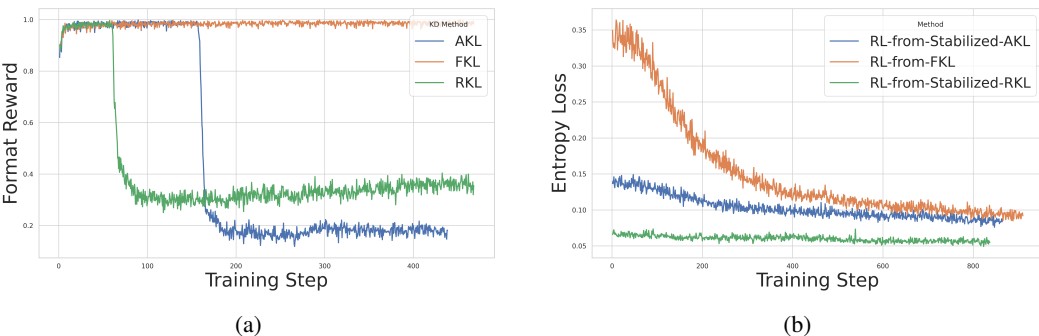

Figure 2: **FKL vs. RKL/AKL.** We compare format rewards and entropy losses with different KL divergences during KD and RL training. **Left:** The RKL/AKL leads to catastrophic training collapse during KD. **Right:** The entropy losses with stablized-RKL/AKL are constantly smaller during RL.

### 3.1.1 INSTABILITY WITH TOP-K TRUNCATION

For computational efficiency, KD is often performed using *top-k truncation*, where the loss is computed only on the teacher's top-$k$ tokens ($V_k(x)$). However, we discover that combining this strategy with the mode-seeking RKL (or its variant AKL) leads to catastrophic training collapse, as shown in Figure 2a. Our analysis shows this is caused by the RKL component, which imposes instable supervision on any token outside $V_k(x)$, destabilizing the optimization. In contrast, top-k FKL remains stable as it simply ignores the tail distribution, imposing no such constraint. A theoretical justification for this instability is provided in Appendix A.3.

### 3.1.2 THE HIDDEN COST OF RKL

Beyond instability, we identify a more fundamental limitation of RKL-based methods: diminished exploratory capacity. Even with a stabilized variant of top-k RKL and its variant AKL

(Appendix A.4), we observe that it consistently yields models that underperform a simple top-k FKL baseline in downstream RL fine-tuning. We attribute this performance deficit to RKL's mode-seeking nature, which aggressively prunes the tail of the student's distribution. While this behavior promotes high-fidelity imitation, it critically reduces the student model's output entropy (Figure 2b), thereby limiting its capacity for exploration—a prerequisite for successful reinforcement learning.

### 3.1.3 OUR APPROACH

These findings motivate our method, **Constrained Knowledge Distillation (CKD)**. We start with the stable and exploration-friendly top-k FKL and introduce a targeted regularization term $\mathcal{L}_{\text{tail}}$ to control the most problematic part of the student's tail distribution. This term $\mathcal{L}_{\text{tail}}$ applies an L1 penalty only to tokens that the student considers probable (in its top-$m$ set, $V_m(x)$) but the teacher deems irrelevant (outside its top-$k$ set, $V_k(x)$).

Our final CKD loss function combines the top-k FKL objective with this targeted tail penalty:

$$\mathcal{L}_{\text{CKD}} = \mathcal{L}_{\text{FKL-k}} + \lambda_{\text{tail}}\mathcal{L}_{\text{tail}} \tag{3}$$

where:

$$\mathcal{L}_{\text{FKL-k}} = \sum_{x\in\mathcal{D}}\sum_{v\in V_k(x)} P_T(v|x)\log\frac{P_T(v|x)}{P_S(v|x)} \tag{4}$$

$$\mathcal{L}_{\text{tail}} = \sum_{x\in\mathcal{D}}\sum_{v\in V_m(x)\setminus V_k(x)} P_S(v|x) \tag{5}$$

and $\lambda_{\text{tail}}$ is a balancing hyperparameter. This approach directly suppresses the student from confidently predicting tokens that the teacher has dismissed. Moreover, according to the detailed gradient analysis (see Appendix A.5), this penalty encourages the redistribution of probability, which implicitly regularizes the student's predictions within the top-k set and discourages over-confidence. It is also beneficial for downstream RL as it retains the capacity for exploration.

## 3.2 SIMILARITY-GUIDED REINFORCEMENT LEARNING (SIM-RL)

Reinforcement Learning with Verifiable Rewards (RLVR) shows significant promise in enhancing the reasoning capabilities of large language models (Lambert et al., 2025). Because the function calling task typically admits multiple valid solutions and meets the challenges of simulating realistic API feedback during training, the reward design often depends on process reward model (PRM) or abstract syntax tree (AST) parsing (Goldie et al., 2025). In this work, we propose Sim-RL, a method that generates reward signals through low-cost computation of similarity between model outputs and ground-truth responses. This approach enables fine-grained similarity-based reward discrimination while effectively mitigating issues of over-rewarding or excessive penalization.

### 3.2.1 REWARD DESIGN

**Format Reward.** A prerequisite for a successful function call is the generation of a response in the correct format. To enable parsing into a structured function call object, the model output must be constrained by a strict format. We illustrate one implementation of format reward using the Qwen tool calling template (see Appendix A.1) as an example. A generation is considered valid if it adheres to the following rules: (1) The output must contain exactly one pair of `<think>...</think>` tags, encapsulating the model's reasoning process; (2) If the model decides to invoke functions, each invocation must be wrapped in `<tool_call>...</tool_call>` tags; (3) The content must be a single JSON object containing two keys: `"name"` and `"arguments"`; (4) The value of the `"name"` key must be present in the set of available functions, $\mathcal{F}$; (5) Furthermore, all keys within the `"arguments"` object must be a subset of the keys defined for that specific function in $\mathcal{F}$.

The format reward $R_{\text{format}}$ is a binary signal defined as:

$$R_{\text{format}} = \begin{cases} 1 & \text{if all format rules are satisfied} \\ 0 & \text{otherwise} \end{cases} \tag{6}$$

**Function Call Reward.** Conditioned on a correct format ($R_{\text{format}} = 1$), we evaluate the accuracy of the tool invocations. Inspired by the Intersection over Union (IoU) principle, the tool call reward compares the predicted sequence of tool calls $P = \{p_1, \ldots, p_m\}$ against the ground-truth sequence $G = \{g_1, \ldots, g_n\}$. It is defined as:

$$R_{\text{fc}} = \frac{\sum_{i=1}^{\min(m,n)} \text{sim}(p_i, g_{\sigma(i)})}{|P| + |G| - |P \cap G|} \tag{7}$$

where $\sigma$ is a greedy matching scheme that establishes a one-to-one correspondence between elements of $P$ and $G$ (see Algorithm 2 in Appendix A.7); $\text{sim}(p, g)$ is an argument-level similarity function between a predicted call $p$ and a ground-truth call $g$:

$$\text{sim}(p, g) = \frac{\sum_{k \in \text{keys}(p) \cap \text{keys}(g)} s(p_k, g_k)}{|\text{keys}(p) \cup \text{keys}(g)|} \tag{8}$$

The function $s(p_k, g_k)$ computes the similarity for a specific argument key $k$, with its definition varying by data type: (1) **String:** ROUGE-L F1 score (Lin, 2004); (2) **Numeric/Boolean:** An exact match (1 if equal, 0 otherwise); (3) **Other types:** An exact match after converting both values to their string representations. Please refer to Algorithm 3 in Appendix A.7.

**Response Reward.** In our task, the model may also generate a natural language response directly without invoking any functions. For such text-only generations, the response reward is defined as the ROUGE-L F1 score between the predicted response $p$ and the ground-truth response $g$:

$$R_{\text{response}} = \text{ROUGE-L}(p, g) \tag{9}$$

**Total Reward.** The total reward $R$ is a composite function that unifies these components:

$$R = \underbrace{(R_{\text{format}} - 1)}_{\text{format term}} + \underbrace{R_{\text{format}} \cdot (R_{\text{fc}} + R_{\text{response}})}_{\text{answer term}} \tag{10}$$

This structure ensures that any format error ($R_{\text{format}} = 0$) results in a strong penalty of -1 from the format term. If and only if the format is correct ($R_{\text{format}} = 1$), the reward transitions to the answer term, evaluating its correctness. The total reward $R$ is thus bounded in the range [-1, 1], guiding the model towards both correct answer formatting and content accuracy. For specific implementation details, please refer to the Appendix A.7.

### 3.2.2 OPTIMIZATION METHOD

We employ GRPO (Shao et al., 2024) as our RL algorithm. GRPO enhances stability by using $G$ rollouts for each prompt and computing the advantage $\hat{A}$ via reward standardization across the group. The objective function is:

$$\mathcal{J}_{\text{GRPO}}(\theta) = \mathbb{E}_{(q,a) \sim \mathcal{D}, \{o_i\}_{i=1}^G \sim \pi_{\theta_{\text{old}}}(\cdot|q)}$$
$$\left[ \frac{1}{G} \sum_{i=1}^{G} \frac{1}{|o_i|} \sum_{t=1}^{|o_i|} \left( \min\left(r_{i,t}(\theta)\hat{A}_{i,t}, \text{clip}(r_{i,t}(\theta), 1-\epsilon, 1+\epsilon)\hat{A}_{i,t}\right) - \beta D_{\text{KL}}(\pi_\theta \| \pi_{\text{ref}}) \right) \right] \tag{11}$$

where the advantage $\hat{A}_{i,t}$ for each token is derived from the standardized reward $R_i$ of its corresponding rollout $o_i$:

$$\hat{A}_{i,t} = \frac{r_i - \text{mean}(\{R_i\}_{i=1}^G)}{\text{std}(\{R_i\}_{i=1}^G)} \tag{12}$$

Given that answer term reward is bounded in [0, 1], a group-wise mean reward of exactly 0 or 1 implies that all rollouts in the group are either entirely incorrect or perfectly correct, respectively. In such cases, the advantage $\hat{A}$ is zero for all samples, so these homogeneous groups contribute no gradient signal. Inspired by DAPO (Yu et al., 2025), we introduce a filtering mechanism to discard these groups from each training batch. This simple yet effective strategy prevents wasted computation and accelerates the RL training process.

### 3.3 THE STAR TRAINING CURRICULUM

Our full training process consists of model distillation and model refinement, as shown in Figure 1.

**Model Distillation:** Effective distillation requires the selection of a good teacher model. A capable, instruction-tuned model (e.g., Qwen3-8B) can serve this purpose. Additionally, inspired by the teacher correction (Sreenivas et al., 2024), we employ the Sim-RL mechanism (see Section 3.2) to better adapt the teacher model to the distillation dataset. Next, we use our stable Constrained Knowledge Distillation (CKD) method to distill the refined teacher's knowledge into the student model. This step effectively transfers the teacher's core capabilities while preventing training instabilities.

**Model Refinement:** Finally, we polish the distilled student's policy with a final application of Sim-RL. This phase corrects minor distillation artifacts and directly optimizes the student's performance and reliability on the most difficult problems.

## 4 EXPERIMENTS

We conduct a series of solid experiments to validate the effectiveness of our STAR methodology.

### 4.1 EXPERIMENTAL SETUP

**Models.** We use the Qwen-family of models (Yang et al., 2025). The teacher model, $\mathcal{M}_T$, is a Qwen3-8B fine-tuned with Sim-RL. The student models, $\mathcal{M}_S$, comprise Qwen3-0.6B, Qwen3-1.7B, and Qwen3-4B, which are trained under the guidance of $\mathcal{M}_T$. Details can be seen in Appendix A.2.

**Datasets.** We construct our initial training set, $\mathcal{D}$, by merging four datasets:

- **ToolACE** (Liu et al., 2025): 11.3k instances of diverse tool usage patterns.
- **xLAM** (Liu et al., 2024): 60k high-quality, validated function calling samples.
- **xLAM-irrelevance** (Lin et al., 2025): 6.7k filtered samples for irrelevant function detection, with answers synthesized using Qwen3-32B.
- **Tool-use-synthetic**[1]: 50k sampled instances of multi-step and multi-turn interactions.

Data in $\mathcal{D}$ is formatted to the Qwen chat specification, with responses validated by a format checker $R_{\text{format}}$. The teacher $\mathcal{M}_T$ then generates rollouts on $\mathcal{D}$ to create an augmented dataset $\mathcal{D}_T$, which includes the teacher's reasoning and final answer. These trajectories are also filtered by $R_{\text{format}}$ to ensure structural correctness. Detailed prompt formats are available in Appendix A.1.

**Baselines.** We compare our method, STAR, against several strong baselines:

- **Base-model**: The pre-trained model without any fine-tuning.
- **SFT**: Standard supervised fine-tuning on the dataset $\mathcal{D}$.
- **SFT-think**: SFT on the teacher-augmented dataset $\mathcal{D}_T$.
- **FKL**: Training on $\mathcal{D}_T$ with a top-k (k=100) forward KL divergence loss, guided by $\mathcal{M}_T$.
- **ToolRL** (Qian et al., 2025): Training the SFT-think model with GRPO and a specialized reward function.
- **LUFFY** (Yan et al., 2025): A hybrid offline-online approach using both $\mathcal{D}$ and $\mathcal{D}_T$ with the Sim-RL reward.
- **GKD** (Agarwal et al., 2024): An online knowledge distillation method trained jointly with RL on $\mathcal{D}$, using the Sim-RL reward and guidance from $\mathcal{M}_T$.

**Benchmarks.** We evaluate all models on two established benchmarks. See details of each evaluation category of benchmarks in Appendix A.9 :

---

[1]`https://huggingface.co/datasets/ai2-adapt-dev/tool-use-synthetic-gpt-4.1-p1`

- **BFCL** (Patil et al., 2025): The de facto standard for function calling evaluation, assessing serial/parallel calls, multi-language support, and multi-step reasoning.
- **ACEBench** (Chen et al., 2025): A new function calling benchmark that enforces a specific output format, challenging a model's instruction-following and generalization abilities.

## 4.2 MAIN RESULTS

Table 1: Performance comparison of different fine-tuning methods on Qwen3-0.6B, evaluated on the BFCLv3 benchmark.

| Method | Overall Acc | Non-Live Acc | Live Acc | Multi Turn Acc |
|---|---|---|---|---|
| *Standard methods* | | | | |
| Base-model | 47.33 | 71.81 | 65.66 | 1.88 |
| SFT | 44.58 | 66.29 | 62.15 | 1.62 |
| SFT-think | 47.59 | 71.54 | 64.46 | 4.50 |
| FKL | 49.51 | 76.44 | 65.93 | 5.12 |
| *Recent methods* | | | | |
| ToolRL | 47.35 | 64.81 | 66.55 | 6.75 |
| LUFFY | 49.23 | 76.75 | 64.59 | 5.48 |
| GKD | 47.32 | 67.62 | 67.61 | 3.25 |
| *Our methods* | | | | |
| CKD | 49.84 | 75.92 | 66.15 | 5.62 |
| Sim-RL | 49.35 | 75.21 | 67.39 | 3.25 |
| SFT+Sim-RL[*] | 50.41 | 76.27 | 66.99 | 6.13 |
| CKD+Sim-RL | **51.70** | **78.65** | **68.19** | **7.00** |

[*]It refers to Sim-RL on SFT-think.
The detailed statistics show in Table 7.

Table 2: Performance comparison of different fine-tuning methods on Qwen3-0.6B, evaluated on the ACEBench Normal benchmark.

| Method | Summary | Atom | Single-Turn | Multi-Turn | Similar API | Preference |
|---|---|---|---|---|---|---|
| *Standard methods* | | | | | | |
| Base-model | 27.20 | 37.70 | 19.50 | 10.00 | 36.00 | 6.00 |
| SFT | 2.10 | 1.70 | 0.50 | 0.00 | 14.00 | 0.00 |
| SFT-think | 28.70 | 42.30 | 14.00 | 9.00 | 34.00 | 10.00 |
| FKL | 36.80 | 52.30 | 16.00 | 16.00 | 42.00 | **22.00** |
| *Recent methods* | | | | | | |
| ToolRL | 29.40 | 45.00 | 12.50 | 10.00 | 34.00 | 4.00 |
| LUFFY | 44.40 | 59.30 | 26.50 | 26.00 | 50.00 | **22.00** |
| GKD | 40.10 | 54.00 | 21.50 | 23.00 | 46.00 | **22.00** |
| *Our methods* | | | | | | |
| CKD | 39.00 | 55.00 | 21.00 | 19.00 | 48.00 | 10.00 |
| Sim-RL | 39.30 | 53.30 | 23.50 | 21.00 | 52.00 | 10.00 |
| SFT+Sim-RL[*] | 38.90 | 53.00 | 21.50 | 21.00 | 46.00 | 18.00 |
| CKD+Sim-RL | **53.00** | **69.30** | **35.00** | **32.00** | **62.00** | 20.00 |

[*]It refers to Sim-RL on SFT-think.
The detailed statistics show in Table 8.

Table 3: Model performance on function calling benchmarks across scales.

| Model | BFCLv3 Overall | ACEBench Normal |
|---|---|---|
| Qwen3-8B | 66.34 | 72.90 |
| Llama3.1-8B | 49.57 | 46.60 |
| Watt-Tool-8B | **67.79** | **75.60** |
| Hammer2.1-7B | 62.25 | 62.80 |
| Teacher-8B | 67.74 | 72.70 |
| Qwen3-4B | 63.39 | 71.80 |
| Llama3.2-3B | 45.86 | 29.60 |
| Hammer2.1-3B | 59.56 | 18.70 |
| STAR-4B | **65.24** | **74.10** |
| Qwen3-1.7B | 54.70 | 51.60 |
| STAR-1.7B | **56.05** | **60.90** |
| Qwen3-0.6B | 47.33 | 27.20 |
| STAR-0.6B | **51.70** | **53.00** |

Table 4: Model performance on function calling benchmarks with different KD strategies.

| Method | BFCLv3 Overall | | ACEBench Normal | |
|---|---|---|---|---|
| | w/o RL | w/ RL | w/o RL | w/ RL |
| CE | 47.59 | 50.41 | 28.70 | 38.90 |
| FKL | 49.51 | 51.46 | 36.80 | 50.00 |
| RSKD | 49.03 | 50.65 | 35.40 | 49.80 |
| RKL* | 49.26 | 50.49 | 35.30 | 41.30 |
| AKL* | 49.47 | 50.29 | **44.20** | 49.00 |
| CKD | **49.56** | **51.70** | 39.00 | **53.00** |

*The stable variant.

**Overall Performance.** As shown in Table 1 and Table 2, our proposed STAR framework, which combines CKD and Sim-RL, establishes a new SOTA for function calling on the 0.6B model scale. On BFCLv3 benchmark, STAR (CKD+Sim-RL) achieves an overall accuracy of **51.70**, and on ACEBench, it scores **53.00**, outperforming all standard and recent methods by a significant margin. Notably, STAR's individual components are also highly effective; CKD and Sim-RL alone surpass most baselines, but their combination yields a synergistic improvement, boosting the BFCLv3 score by over 2 points and the ACEBench score by 14 points compared to their individual applications. The detailed statistics show in Appendix A.10 and the additional results show in Appendix A.12.

**Superior Generalization and Robustness.** STAR's generalization capabilities are a key advantage of our framework. Standard Supervised Fine-Tuning (SFT) leads to a performance collapse on ACEBench, as the model severely overfits to the JSON format of the training data and fails to adapt to the benchmark's Python-style function call syntax. In stark contrast, the STAR-trained model, despite being trained on the same data, demonstrates exceptional robustness. It successfully generalizes its learned function calling abilities to the unseen format, highlighting that our KD+RL paradigm teaches the model underlying reasoning rather than mere format mimicry.

**Performance Across Scales.** We validate the effectiveness of STAR across various model sizes, as detailed in Table 3. Our STAR-trained models consistently outperform their base model counterparts and other models of similar scale. The results demonstrate that STAR significantly closes the performance gap with much larger models. For instance, our STAR-0.6B model (53.00 on ACEBench) substantially surpasses the much larger Llama3.1-8B (46.60). And our STAR-4B (74.10) outperforms Qwen3-8B (72.90) on ACEBench. This showcases the framework's potent ability to distill and refine capabilities into smaller, more efficient models across various scales.

## 4.3 ANALYSIS

**Why KD+RL over SFT+RL for Super-Tiny Models?** The prevalent SFT+RL paradigm, while effective for large models, proves suboptimal for super-tiny models. SFT's hard supervision forces small, limited-capacity models to overfit and "memorize" specific output formats . This leads to a policy with limited generalization, as evidenced by its failure and low Pass@k score on ACEBench (Figure 3 and Table 2), and creates a poor initialization for RL that limits refinement potential. In contrast, our STAR framework forces the student to mimic the teacher's full probability distribution using "soft" supervision through KD training. This encourages learning the teacher's reasoning and uncertainty, resulting in a more robust and generalizable initial policy as a stronger foundation for the subsequent Sim-RL refinement.

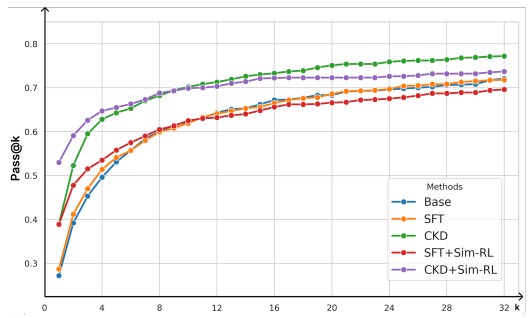
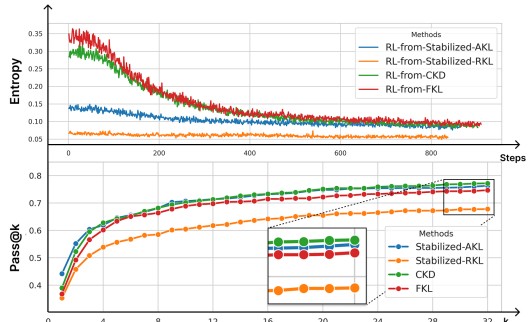

Figure 3: Comparison of Pass@k performance for different methods.

Figure 4: Comparison of Pass@k performance and entropy for different KD methods.

**The Role of Constrained Distillation.** Our ablation over various KD strategies (Table 4) justifies choosing CKD. While all KD methods, including recent approaches like RSKD (Anshumann et al., 2025), are better initializers for Sim-RL than cross-entropy (CE), CKD consistently yields the best final performance. Crucially, the CKD-initialized policy already exhibits superior reasoning capacity before RL, achieving the highest Pass@k scores among all initializers (Figure 4, bottom). This metric is a vital indicator of a model's potential, measuring its ability to generate a diverse set of correct solutions rather than relying on a single, high-confidence prediction (Deng et al., 2025; Kang et al., 2025). This advantage stems from CKD's unique re-balancing of learning signals: it preserves the teacher's top-k probabilities while introducing a targeted suppression term that penalizes "confident-but-wrong" logits more forcefully. This focused distillation creates a superior policy initialization that is more amenable to RL refinement by endowing the model with significantly higher policy entropy at the start of RL training (Figure 4, top). Such entropy is essential for effective exploration and preventing premature convergence in RL (Sutton, 1988; Cui et al., 2025b). This approach synergizes most effectively with Sim-RL, because CKD achieves higher Pass@k and policy entropy than other advanced methods like Stabilized AKL, which are limited by a suppressed policy entropy that prevents their gains from translating well post-RL. It underscores the importance of our constrained approach.

**Similarity-based Reward Design.** Our ablation over reward designs (Table 5) demonstrates the inadequacy of standard metrics, like binary reward (Hao et al., 2025), for complex tasks. A binary reward proves brittle, failing to generalize as it harshly penalizes functionally correct yet syntactically varied solutions. While more advanced methods like the specialized ToolRL reward and SWiRL (Goldie et al., 2025), a Process Reward Model (PRM) based variant, offer improvements, our Sim-RL consistently achieves superior performance, especially on the challenging generalization benchmark. This advantage stems from its fine-grained, continuous reward signal, which evaluates output similarity rather than a strict pass/fail criterion. This richer signal more effectively guides the policy towards a diverse set of valid solutions, enhancing generalization and confirming that a task-aligned similarity metric is crucial for optimal policy refinement. The case study shows in Appendix A.11.

Table 5: Ablation study on reward designs.

| Method | BFCLv3 Overall | ACEBench Normal |
|---|---|---|
| CKD+Binary Reward | 51.05 | 35.70 |
| CKD+ToolRL | 48.59 | _40.50_ |
| CKD+SwiRL | _51.10_ | 40.30 |
| CKD+Sim-RL | **51.70** | **53.00** |

## 5 RELATED WORKS

**LLM for Function Calling**    Function calling is a fundamental capability for agentic AI, enabling models to interact with external tools. Early research showed that self-supervised learning could improve zero-shot tool-calling capabilities (Schick et al., 2023). Subsequently, a dominant paradigm has been supervised fine-tuning (SFT) on large-scale, synthetically generated datasets with verifiable tool calls (Liu et al., 2024; 2025; Li et al., 2023). To mitigate impaired generalization caused by naive SFT, researchers have introduced strategies like masking (Lin et al., 2025). More recently, reinforcement learning (RL) has been applied on top of SFT to further enhance performance (Qian et al., 2025). Notably, these advancements are not exclusive to large models, as targeted training has enabled even 1B-scale models to achieve practical tasks like web browsing (Erdogan et al., 2024).

**Knowledge Distillation**    Knowledge Distillation (KD) trains a compact student model to mimic a larger teacher, originally by matching its output probability distribution (Hinton et al., 2015). Prevailing methods distill knowledge from the teacher's output logits (Gu et al., 2024; Kim et al., 2024), intermediate features (Yang et al., 2023), or entire sequences (Kim & Rush, 2016). Logits-based approaches, which are common, typically minimize the forward KL divergence (FKL) (Sanh et al., 2020; Kim et al., 2023), reverse KL divergence (RKL) (Gu et al., 2024; Li et al., 2024), or both (Wu et al., 2025). More recently, top-k distillation has been explored to improve computational and storage efficiency (Anshumann et al., 2025; Peng et al., 2025).

**Reinforcement Learning**    Enhancing the reasoning abilities of Large Language Models (LLMs) through RL has emerged as a prominent research direction (Hu et al., 2025b; Xie et al., 2025; Pan et al., 2025). This line of inquiry has yielded several high-performing models, including DeepSeek-R1 (DeepSeek-AI et al., 2025), Qwen3 (Yang et al., 2025), and OpenAI's o1 (Jaech et al., 2024). Central to these advancements is Proximal Policy Optimization (PPO) (Schulman et al., 2017), a foundational RL algorithm. Building on PPO, Group Relative Policy Optimization (GRPO) (Shao et al., 2024) simplifies the training pipeline by incorporating verifiable rule-based rewards. Subsequently, DAPO (Yu et al., 2025) further refines GRPO with techniques like clip-higher and dynamic sampling, boosting both training efficiency and performance. In parallel, SFT is now standard practice for initializing RL training (Cui et al., 2025a), motivating further research into hybrid paradigms that optimize the synergy between SFT and RL (Yan et al., 2025; Ma et al., 2025).

## 6 CONCLUSION

We introduce STAR, a framework combining constrained knowledge distillation (CKD) and a similarity-driven RL mechanism Sim-RL to transfer LLMs' capabilities to super-tiny models for efficient, low-latency deployment. Empirically, STAR establishes a new performance benchmark for this model class, rivaling and even surpassing some larger models. Our analysis demonstrates that our training curriculum is superior to conventional paradigms for low-capacity models, effectively transferring teacher competence into a generalizable student policy. We position STAR as a promising approach for principled small-model specialization. We hope this work catalyzes further research on compact, reliable agents—exploring multi-teacher strategies, richer reward designs, and deployment-aware constraints—to make capable models accessible where they are most needed.

## 7 LIMITATION & FUTURE WORK

While STAR demonstrates strong performance on function calling, several limitations warrant further investigation. First, our current work is validated on function calling, yet the underlying framework shows promising potential for generalization to other tasks (e.g., SQL generation, mathematical reasoning), which presents a promising avenue for future work. Second, we have explored some similarity-guided rewards to improve the training process. While this initial approach has proven effective, a more comprehensive investigation into alternative and potentially more sophisticated similarity measures is left for future work. Such an exploration could help in designing more granular feedback, although the potential performance gains remain to be quantified.

ETHICS STATEMENT

Our research is conducted in full alignment with the ICLR Code of Ethics. This study did not involve any human participants or animal experimentation. The datasets employed, namely ToolACE, xLAM, xLAM-irrelevance, and Tool-use-synthetic, were all procured in strict accordance with their respective usage policies, ensuring no infringement on privacy. Throughout our methodology, we have diligently worked to prevent the introduction of bias and to avoid producing discriminatory results. Furthermore, no personally identifiable information (PII) was processed, and our experimental procedures were designed to pose no risks to privacy or security. We uphold a steadfast commitment to the principles of transparency and integrity in all aspects of our work.

REPRODUCIBILITY STATEMENT

To ensure the reproducibility of our work on the STAR framework, we provide detailed descriptions of our methodology and experimental setup. Our core methods, Similarity-guided Reinforcement Learning (Sim-RL) and Constrained Knowledge Distillation (CKD), are described in Section 3.2 and Section 3.1. The precise algorithm for our novel similarity-based reward function is detailed via pseudo-code in Appendix A.7. Our full experimental setup, including the specific models, datasets, and benchmarks used, is outlined in Section 4.1. All training hyperparameters, such as learning rates, batch sizes, and method-specific constants, are comprehensively listed in Appendix A.2. Furthermore, details on the prompt formats used for evaluation are available in Appendix A.1. The code is open-sourced and accessible at `https://github.com/Qwen-Applications/STAR`.

ACKNOWLEDGMENTS

We thank the anonymous reviewers and the area chair for their valuable feedback, which significantly improved this paper. We also extend our appreciation to Dakui Wang and Xin Li for their generous assistance with the research infrastructure.

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

# A APPENDIX

## A.1 PROMPT

Our training data is organized according to the Qwen chat template. On BFCL, we employed the QwenHandler with a customized system prompt (see Figure 5). Conversely, to adhere to the strict evaluation protocol of ACEBench, we used its official, unmodified prompt template[2].

## A.2 TRAINING DETAILS

All experiments were conducted using the OpenRLHF framework (Hu et al., 2025a) on a single server equipped with 8 NVIDIA H20 GPUs. For the various training schemes in our experiments, we employed the following hyperparameter settings:

- **Reinforcement Learning (RL)**: For RL training, we employed GRPO for fine-tuning. We set a constant learning rate of 3e-7, with both rollout and training batch sizes of 128. The KL-divergence constraint was managed via the k2 approximation, with an initial KL coefficient of 1e-3. For each prompt, 8 response rollouts were generated.

- **Knowledge Distillation (KD)**: For KD training, the model was optimized with a learning rate of 3e-6 and a batch size of 128. For the tail-suppression term, $\lambda_{\text{tail}}$ was set to 10, and both k and m were fixed at 100.

- **Supervised Fine-tuning(SFT)**: For SFT, the learning rate is fixed at 2e-5, while the batch size is set to 128.

## A.3 ANALYSIS OF TOP-K FKL AND RKL

This work dissects the gradient dynamics of top-k knowledge distillation to provide a principled explanation for the contrasting empirical performance of Forward KL (FKL) and Reverse KL (RKL) divergences. We reveal that FKL's success stems from a stable, bounded gradient, whereas RKL is prone to instability due to a potentially unbounded gradient signal, thereby elucidating the fundamental mechanism behind their differing behaviors.

**Notation:**

- A teacher model produces logits $z_T \in \mathbb{R}^C$, yielding a probability distribution $p = \text{softmax}(z_T)$.

- A student model produces logits $z_S \in \mathbb{R}^C$, yielding a probability distribution $q = \text{softmax}(z_S)$.

- We denote $I_k = \text{top-k-indices}(p)$ as the index set of the the $k$ largest probabilities in the teacher distribution $p$. As this set is determined solely by the teacher, it is treated as a constant in the gradient computation with respect to the student's parameters.

### A.3.1 ANALYSIS OF TOP-K FKL

The top-K FKL loss is defined as:

$$\mathcal{L}_{\text{FKL-TopK}} = \sum_{i \in I_k} p_i \log \frac{p_i}{q_i} = \sum_{i \in I_k} p_i (\log p_i - \log q_i) \tag{13}$$

---

[2]The official ACEBench prompt is available at: `https://github.com/chenchen0103/ACEBench/blob/main/model_inference/prompt_en.py`

```
<|im_start|>assistant
# You are a helpful assistant.  The assistant first thinks
about the reasoning process in the mind and then provides
the user with the answer.  The reasoning process are
enclosed within <think>explain why the user's question can
be answered without calling a function or why you should
ask the user for more information or why you should call
one or more functions and your plan to solve the user's
question.</think>, and then give the answer.  You can call
the tool by <tool_call> </tool_call> tag.
# If the user's question can be answered without calling any
function, please answer the user's question directly.  In
this situation, you should return your thought and answer
the user's question directly.
# If the user cannot be answered without calling any
function, and the user does not provide enough information
to call functions, please ask the user for more information.
In this situation, you should return your thought and ask
the user for more information.
# If the user's question cannot be answered without
calling any function, and the user has provided enough
information to call functions to solve it, you should call
the functions.  In this situation, the assistant should
return your thought and call the functions.
# Tools
You may call one or more functions to assist with the user
query.
You are provided with function signatures within
<tools></tools> XML tags:
<tools>
{"name":  "earnings.getbymonth", "description":
"Fetches earning data for a specific month and year
using the RapidAPI service.", "parameters":  {"month":
{"description":  "The month for which to fetch earnings
data.", "type":  "str", "default":  "05"}, "year":
{"description":  "The year for which to fetch earnings
data.", "type":  "str", "default":  "2022"}}} {"name":
"creditcard.generate_cc_number", "description":  "Generates
a fake credit card number using the specified brand and
API key.", "parameters":  {"brand":  {"description":  "The
desired card brand (e.g., 'Visa', 'MasterCard').  Defaults
to None.", "type":  "str, optional", "default":  ""}}}
</tools>
For each function call, return a json object with function
name and arguments within <tool_call></tool_call> XML tags:
<tool_call>
{"name":  <function-name>, "arguments":  <args-json-object>}
</tool_call><|im_end|>
<|im_start|>user
I want to analyze the market for Apple (AAPL). First, give
me the most recent Minus Directional Indicator (MINUS_DI) for
AAPL using daily intervals, then interpret what that value
implies for the stock's short-term price movement.<|im_end|>
<|im_start|>
assistant
<think>
```

Figure 5: Customized system prompt example on BFCL evaluation.

The partial derivative of the loss with respect to a student logit $z_{S_j}$ is found by applying the chain rule with the softmax derivative $\frac{\partial q_i}{\partial z_{S_j}} = q_i(\delta_{ij} - q_j)$, yielding:

$$
\begin{aligned}
\frac{\partial \mathcal{L}_{\text{FKL-TopK}}}{\partial z_{S_j}} &= \sum_{i=1}^{C} \frac{\partial \mathcal{L}}{\partial q_i} \frac{\partial q_i}{\partial z_{S_j}} = \sum_{i \in I_k} \left( -\frac{p_i}{q_i} \right) \frac{\partial q_i}{\partial z_{S_j}} \\
&= \sum_{i \in I_k} \left( -\frac{p_i}{q_i} \right) q_i(\delta_{ij} - q_j) \\
&= q_j \sum_{i \in I_k} p_i - p_j \cdot \mathbf{1}_{j \in I_k}
\end{aligned}
\tag{14}
$$

To elucidate the underlying training dynamics, we decompose the FKL-TopK gradient by analyzing its components for logits within the top-k set versus non-top-k set:

**For a non-top-k logit ($j \notin I_k$):**

$$
\frac{\partial \mathcal{L}_{\text{FKL-TopK}}}{\partial z_{S_j}} = q_j \sum_{i \in I_k} p_i
\tag{15}
$$

**For a top-k logit ($j \in I_k$):**

$$
\frac{\partial \mathcal{L}_{\text{FKL-TopK}}}{\partial z_{S_j}} = q_j \sum_{i \in I_k} p_i - p_j
\tag{16}
$$

This formulation induces a learning dynamic where logits for top-k and non-top-k items receive fundamentally different treatments. The gradient for a top-k logit is strictly smaller than the gradient for any non-top-k logit (since $p_j > 0$). This creates a clear, stable dynamic: the logits of non-top-k items are strongly suppressed, while the logits of top-K items are either encouraged (if the gradient is negative) or suppressed much more weakly. The model learns to focus its probability mass on the teacher's chosen top-K candidates.

### A.3.2 ANALYSIS OF TOP-K RKL

The RKL objective presents a fundamental issue. If we define a proper probability distribution $p'$ from the teacher's top-k logits by padding with zeros (i.e., $p'_i = 0$ for $i \notin I_k$), the RKL $D_{KL}(q||p')$ becomes ill-defined. Any student probability $q_i > 0$ for an index $i \notin I_k$ would result in a term $q_i \log(q_i/0)$, causing the loss to diverge to infinity, which is impossible to optimize.

The only viable alternative is a masked RKL, which is not a true KL divergence over the full vocabulary:

$$
\mathcal{L}_{\text{RKL-TopK}} = \sum_{i \in I_k} q_i \log \frac{q_i}{p_i}
\tag{17}
$$

The gradient of this loss with respect to a student logit $z_{S_j}$ is:

$$
\begin{aligned}
\frac{\partial \mathcal{L}_{\text{RKL-TopK}}}{\partial z_{S_j}} &= \sum_{i \in I_k} \frac{\partial (q_i \log \frac{q_i}{p_i})}{\partial q_i} \frac{\partial q_i}{\partial z_{S_j}} \\
&= \sum_{i \in I_k} \left( \log \frac{q_i}{p_i} + 1 \right) q_i(\delta_{ij} - q_j) \\
&= q_j \left[ \left( \log \frac{q_j}{p_j} + 1 \right) \mathbf{1}_{j \in I_k} - \sum_{i \in I_k} q_i \left( \log \frac{q_i}{p_i} + 1 \right) \right]
\end{aligned}
\tag{18}
$$

Let's analyze the dynamics by defining the summation term $S = \sum_{i \in I_k} q_i(\log \frac{q_i}{p_i} + 1)$.

**For a non-top-k logit ($j \notin I_k$):**

$$
\frac{\partial \mathcal{L}_{\text{RKL-TopK}}}{\partial z_{S_j}} = -q_j S
\tag{19}
$$

**For a top-k logit ($j \in I_k$):**

$$\frac{\partial \mathcal{L}_{\text{RKL-TopK}}}{\partial z_{S_j}} = q_j \left( \log \frac{q_j}{p_j} + 1 - S \right) \tag{20}$$

This structure, however, can induce undesirable optimization dynamics. Specifically, when the teacher assigns negligible probabilities ($p_j \to 0$) to certain top-k items, or when the student becomes over-confident (i.e., its probability mass $q_j$ is highly concentrated), the gradients for some top-k logits can become smaller than those for non-top-k logits. In this regime, the model is paradoxically incentivized to promote non-top-k items over some within the top-k set, irrespective of an external signal $S$. This behavior often leads to poor convergence and, in extreme cases, training collapse.

## A.4 STABLE VARIANT OF TOP-K RKL AND AKL

To remedy the instability of top-k RKL and Adaptive KL divergence (AKL), we introduce a tail suppression term, analogous to the one used in CKD. Let $J_m = \text{top-m-indices}(q)$ be the indices of the student's top-m predictions, and $J'_m = J_m \setminus I_k$ be the set of "confident but wrong" predictions. The stabilized top-K RKL loss is defined as:

$$\mathcal{L}_{\text{Stabilized-RKL-TopK}} = \mathcal{L}_{\text{RKL-TopK}} + \mathcal{L}_{tail} = \sum_{i \in I_k} q_i \log \frac{q_i}{p_i} + \sum_{j \in J'_m} q_i \tag{21}$$

The gradient of $L_{tail}$ with respect to a student logit $z_{S_j}$ is:

$$\begin{aligned}
\frac{\partial \mathcal{L}}{\partial z_{S_j}} &= \sum_{i \in J'_m} \frac{\partial(\lambda q_i)}{\partial q_i} \frac{\partial q_i}{\partial z_{S_j}} \\
&= \sum_{i \in J'_m} \lambda q_i (\delta_{ij} - q_j) \\
&= \lambda q_j \cdot \mathbf{1}_{j \in J'_m} - \lambda q_j \sum_{i \in J'_m} q_i
\end{aligned} \tag{22}$$

Let's analyze the the new gradients with the tail suppression term.

**For a top-k logit ($j \in I_k$):**

$$\frac{\partial \mathcal{L}_{\text{Stabilized-RKL-TopK}}}{\partial z_{S_j}} = q_j \left[ \log \frac{q_j}{p_j} + 1 - S + \lambda \left( 1 - \sum_{i \in J'_m} q_i \right) \right] \tag{23}$$

**For a confident-but-wrong logit ($j \in J'_m$):**

$$\frac{\partial \mathcal{L}_{\text{Stabilized-RKL-TopK}}}{\partial z_{S_j}} = q_j \left[ \lambda \left( 1 - \sum_{i \in J'_m} q_i \right) - S \right] \tag{24}$$

The tail suppression term introduces a positive component $\lambda q_j (1 - \sum_{i \in J'_m} q_i)$. For a sufficiently large $\lambda$, the gradient for a confident-but-wrong logit is larger than that for a top-k logit. This restores a stable learning dynamic by ensuring that the student is penalized for confidently predicting classes outside the teacher's top-k set.

## A.5 GRADIENT ANALYSIS FOR CKD

Our proposed CKD method combines top-k FKL with the same tail suppression mechanism (see Equation 3). Unlike with RKL, the goal here is not to fix instability but to refine the already stable FKL dynamics to prevent over-confidence. The gradient with respect to $z_{S_j}$ is:

**For a top-k logit** $(j \in I_k)$**:**

$$\frac{\partial \mathcal{L}_{\text{CKD}}}{\partial z_{S_j}} = q_j \left( \sum_{i \in I_k} p_i - \lambda \sum_{i \in J'_m} q_i \right) - p_j \tag{25}$$

**For a confident-but-wrong logit** $(j \in J'_m)$**:**

$$\frac{\partial \mathcal{L}_{\text{CKD}}}{\partial z_{S_j}} = q_j \left[ \sum_{i \in I_k} p_i + \lambda \left( 1 - \sum_{i \in J'_m} q_i \right) \right] \tag{26}$$

**For other non-top-k logits** $(j \notin I_k \cup J'_m)$**:**

$$\frac{\partial \mathcal{L}_{\text{CKD}}}{\partial z_{S_j}} = q_j \left( \sum_{i \in I_k} p_i - \lambda \sum_{i \in J'_m} q_i \right) \tag{27}$$

Compared to the standard FKL gradient in Equation 14, CKD strategically re-balances the learning signals:

1. **Targeted Suppression:** The gradient for "confident-but-wrong" logits $(j \in J'_m)$ is significantly increased. This focuses the suppressive force on the most likely sources of error, penalizing the student for being confident in incorrect predictions.
2. **Relaxed Suppression:** The gradient for other non-top-k logits $(j \notin I_k \cup J'_m)$ is reduced. This tells the model not to waste capacity aggressively suppressing classes it already assigns low probability to.

This re-balancing mechanism prevents the model from collapsing its probability mass entirely onto the top-k set $I_k$. By forcing the student to specifically avoid confident mistakes outside of $I_k$, CKD encourages a healthier, less peaky student distribution, which translates to improved generalization and robustness, thus addressing the primary limitation of top-k FKL.

## A.6  SENSITIVITY ANALYSIS

Table 6: Sensitivity analysis on hyperparameters $k$ and $\lambda_{tail}$ for CKD.

| $k$ | $\lambda_{tail}$ | BFCL v3 Overall | | AceBench Normal | |
|---|---|---|---|---|---|
| | | w/o RL | w/ RL | w/o RL | w/ RL |
| 10 | | 49.58 | 51.48 | 43.20 | 49.20 |
| 100 | 10 | 49.56 | 51.70 | 39.00 | 53.00 |
| 1000 | | 49.84 | 51.59 | 36.70 | 52.20 |
| | 1 | 50.12 | 51.11 | 38.00 | 48.20 |
| | 3 | 48.78 | 50.62 | 39.10 | 50.10 |
| 100 | 10 | 49.56 | 51.70 | 39.00 | 53.00 |
| | 30 | 49.82 | 51.80 | 41.70 | 47.80 |
| | 100 | 48.85 | 51.83 | 42.10 | 48.20 |

We investigate the sensitivity of our Constrained Knowledge Distillation (CKD) method to its two key hyperparameters: $k$ and $\lambda_{\text{tail}}$. Table 6 presents the results on both BFCLv3 and ACEBench-Normal benchmarks, with and without subsequent Sim-RL refinement. For this analysis, $m$ is fixed at 100, a value large enough to capture the student's most probable and potentially erroneous outputs.

First, we observe that across a wide range of hyperparameter settings, CKD maintains strong performance, frequently surpassing the results of competing methods shown in Tables 1 and 2. This demonstrates the robustness of our proposed approach.

**Analysis of $k$.** The hyperparameter $k$ defines the size of the trusted vocabulary set from the teacher model. A very small $k$ (e.g., $k = 10$) overly constrains the student, forcing it to mimic a narrow distribution, which can harm generalization as reflected by the lower performance on ACEBench post-RL. Conversely, a very large $k$ (e.g., $k = 1000$) makes the $\mathcal{L}_{\text{FKL-k}}$ term approximate the standard forward KL divergence and reduces the impact of the tail penalty, offering diminishing returns while still performing well. Our chosen value of $k = 100$ strikes an effective balance, providing sufficient guidance from the teacher without excessively restricting the student's distribution, proving beneficial for both initial distillation and subsequent RL adaptation.

**Analysis of $\lambda_{\text{tail}}$.** The weight $\lambda_{\text{tail}}$ controls the strength of the tail suppression penalty. A small weight (e.g., $\lambda_{\text{tail}} = 1$) is insufficient to suppress the student's tendency to assign probability to irrelevant tokens, leading to suboptimal performance after RL. As $\lambda_{\text{tail}}$ increases, performance improves, peaking at $\lambda_{\text{tail}} = 10$, especially on ACEBench. However, excessively large values (e.g., $\lambda_{\text{tail}} \geq 30$) can be overly punitive. This may excessively suppress the student's output probabilities, making the distribution too sharp and hindering the exploratory capacity that is crucial for effective RL fine-tuning, as reflected by the performance drop on ACEBench. Thus, $\lambda_{\text{tail}} = 10$ provides an optimal balance that effectively regularizes the tail distribution while preserving a healthy capacity for exploration.

## A.7 PESUDO CODE OF THE REWARD

The total reward $R$ is calculated using a composite function that first evaluates the syntactic format and then, if the format is correct, the accuracy of the tool calls or the textual response. The framework is defined by the main function `CalculateTotalReward`(see Algorithm 1 and its subroutines.

---

**Algorithm 1** Total Reward Calculation

---

1: **function** CALCULATETOTALREWARD(Generation, GroundTruth, ToolSchema)
2:     **Input:**
3:         *Generation*: The full string output from the model.
4:         *GroundTruth*: The label string containing the correct output.
5:         *ToolSchema*: A definition of available tools $\mathcal{F}$ and their parameters.
6:     **Output:**
7:         $R$: The final reward score in the range [-1, 1].
8:     $R_{\text{format}} \leftarrow$ CALCULATEFORMATREWARD($Generation, ToolSchema$)
9:     **if** $R_{\text{format}} = 0$ **then**
10:         **return** -1
11:     **end if**
12:     $P_{\text{calls}}, P_{\text{response}} \leftarrow$ PARSE($Generation$)
13:     $G_{\text{calls}}, G_{\text{response}} \leftarrow$ PARSE($GroundTruth$)
14:     $R_{\text{tool}} \leftarrow 0$
15:     $R_{\text{response}} \leftarrow 0$
16:     **if** $G_{\text{calls}}$ is not empty **then**
17:         $R_{\text{tool}} \leftarrow$ CALCULATETOOLREWARD($P_{\text{calls}}, G_{\text{calls}}$)         ▷ See Algorithm 2
18:     **else**
19:         $R_{\text{response}} \leftarrow$ CALCULATERESPONSEREWARD($P_{\text{response}}, G_{\text{response}}$)
20:     **end if**
21:     $R \leftarrow R_{\text{tool}} + R_{\text{response}}$
22:     **return** $R$
23: **end function**

---

## A.8 THE USE OF LARGE LANGUAGE MODELS (LLMS)

During the preparation of this manuscript, we employed a Large Language Model (LLM), specifically Gemini 2.5 Pro as a tool for linguistic enhancement and technical formatting, to assist in polishing the language in certain sections. The model's contributions were directed towards improv-

---

**Algorithm 2** Tool Call Reward Calculation ($R_{\text{tool}}$)

---

1: **function** CALCULATETOOLREWARD($P_{\text{calls}}, G_{\text{calls}}$)
2:      total_similarity $\leftarrow 0$
3:      $G'_{\text{calls}} \leftarrow$ a mutable copy of $G_{\text{calls}}$
4:      **for** each predicted call $p \in P_{\text{calls}}$ **do**
5:          best_match_score $\leftarrow -1$
6:          best_match_g $\leftarrow$ null
7:          **for** each ground-truth call $g \in G'_{\text{calls}}$ **do**
8:              **if** $p$.name $= g$.name **then**
9:                  $s \leftarrow$ ARGUMENTSIMILARITY($p$.arguments, $g$.arguments)      ▷ See Algorithm 3
10:                  **if** $s >$ best_match_score **then**
11:                      best_match_score $\leftarrow s$
12:                      best_match_g $\leftarrow g$
13:                  **end if**
14:              **end if**
15:          **end for**
16:          **if** best_match_g is not null **then**
17:              total_similarity $\leftarrow$ total_similarity $+$ best_match_score
18:              Remove best_match_g from $G'_{\text{calls}}$
19:          **end if**
20:      **end for**
21:      union_size $\leftarrow |P_{\text{calls}}| + |G_{\text{calls}}|$
22:      **if** union_size $= 0$ **then return** 1
23:      **elsereturn** total_similarity$/$union_size
24:      **end if**
25: **end function**

---

**Algorithm 3** Argument-level Similarity (sim)

---

1: **function** ARGUMENTSIMILARITY($P_{\text{args}}, G_{\text{args}}$)
2:      intersection_keys $\leftarrow$ keys($P_{\text{args}}$) $\cap$ keys($G_{\text{args}}$)
3:      union_keys $\leftarrow$ keys($P_{\text{args}}$) $\cup$ keys($G_{\text{args}}$)
4:      weighted_sum $\leftarrow 0$
5:      **for** each key $k \in$ intersection_keys **do**
6:          $p_k \leftarrow P_{\text{args}}[k]$, $g_k \leftarrow G_{\text{args}}[k]$
7:          **if** $p_k, g_k$ are Strings **then**
8:              score $\leftarrow$ ROUGE-L_F1($p_k, g_k$)
9:          **else if** $p_k, g_k$ are Numeric/Boolean **then**
10:              score $\leftarrow$ (1 if $p_k = g_k$ else 0)
11:          **else**
12:              score $\leftarrow$ (1 if str($p_k$) $=$ str($g_k$) else 0)
13:          **end if**
14:          weighted_sum $\leftarrow$ weighted_sum $+$ score
15:      **end for**
16:      **if** $|$union_keys$| = 0$ **then return** 1
17:      **else return** weighted_sum$/|$union_keys$|$
18:      **end if**
19: **end function**

---

ing readability, clarity, and the overall flow of the text through tasks such as sentence rephrasing and grammar correction. Furthermore, it was used for assistance with LaTeX syntax to ensure proper formatting.

It must be emphasized that the LLM's function was strictly supportive and limited to the aspects mentioned above. The model had no role in the core intellectual work, which includes the ideation, research methodology, experimental design, data analysis, or interpretation of results. All scientific ideas, concepts, and analyses presented herein are exclusively conducted by the authors.

The authors have thoroughly reviewed and verified the entire manuscript and take full responsibility for its final content, including all text polished or formatted with the aid of the LLM.

## A.9 DETAILS OF EVALUATION METRICS

**Evaluation Metrics for BFCLv3.** The evaluation metrics for BFCLv3 are listed below:

- **Overall Acc**: This metric represents the comprehensive performance of the model on the entire BFCLV3 benchmark. It is calculated as a weighted average of the accuracies from various specific evaluation categories, providing a single, overarching score to rank different methods.

- **Non-Live Acc**: This metric assesses model performance primarily on the static BFCL V1 dataset. This dataset was curated by the benchmark creators and includes single-turn scenarios like simple, multiple, and parallel function calls. As noted in the documentation, this portion of the benchmark may be susceptible to data contamination for models trained on public datasets.

- **Live Acc**: This metric measures model performance on the BFCL V2 live dataset. This dataset is composed of live, user-contributed function documentation and queries, designed to tackle issues of data contamination and bias. It aims to faithfully evaluate a model's ability to generalize and perform effectively in diverse, real-world tool-use scenarios that it has not seen before.

- **Multi Turn Acc**: Introduced with the BFCL V3 dataset, this metric specifically evaluates the model's proficiency in handling multi-turn and multi-step function calling tasks. It tests the model's ability to maintain conversational context over several exchanges, correctly interpret user follow-up requests, and make appropriate function calls based on the accumulated dialogue history.

**Evaluation Metrics for ACEBench-Normal.** The evaluation metrics for ACEBench-Normal are listed below:

- **Summary**: This is a summary score that aggregates the performance across all subcategories within the Normal dataset to provide a single, comprehensive measure of the model's general tool-use capability in standard scenarios.

- **Atom**: This metric evaluates the model's performance on atomic cases, with a specific focus on its ability to handle different parameter types. It involves the precise assessment of the model's handling of data types such as enums, numbers, lists, booleans, and objects.

- **Single-Turn**: This metric assesses the model's basic tool-calling competence in scenarios that are resolved within a single conversational turn.

- **Multi-Turn**: This metric measures the model's capability in multi-turn dialogue flows. It assesses whether the model can perform context-sensitive orchestration of tool calls and maintain state memory across several conversational turns to fulfill the user's goal.

- **Similar API**: This metric tests the model's ability to distinguish between nearly identical tool specifications. The model must select the correct API based on subtle differences in the user's query and the API documentation.

- **Preference**: This metric evaluates if the model can incorporate contextual user information for API selection. The model must make a preference-based selection by taking the user's history or profile into account.

## A.10 STATISTICAL SIGNIFICANCE

To ensure the statistical significance of our findings, we conduct experiments with 3 different random seeds. The results are presented as mean ± standard deviation. Tables 7 and 8 detail the comprehensive performance of our methods against various baselines on the BFCLv3 and ACEBench Normal benchmarks, respectively, showing that our CKD+Sim-RL approach consistently achieves strong results across most evaluation metrics. We further validate our architectural choices through ablation studies: Table 9 confirms the superiority of our proposed CKD strategy over other knowledge distillation techniques, and Table 10 demonstrates the significant advantage of our Sim-RL reward design

compared to alternative reward functions. The consistently low variance and superior performance of our proposed components across all these detailed tables underscore that our method is robust and its effectiveness is statistically well-supported.

Table 7: Performance comparison of different fine-tuning methods on Qwen3-0.6B, evaluated on the BFCLv3 benchmark. Values are presented as mean $\pm$ std.

| Method | Overall Acc | Non-Live Acc | Live Acc | Multi Turn Acc |
|---|---|---|---|---|
| *Standard methods* | | | | |
| Base-model[†] | 47.33 | 71.81 | 65.66 | 1.88 |
| SFT | $44.19 \pm 0.17$ | $65.40 \pm 0.33$ | $61.67 \pm 0.05$ | $1.45 \pm 0.15$ |
| SFT-think | $47.14 \pm 0.20$ | $69.82 \pm 0.81$ | $64.59 \pm 0.33$ | $3.94 \pm 0.31$ |
| FKL | $49.68 \pm 0.12$ | $76.05 \pm 0.26$ | $66.16 \pm 0.41$ | $5.41 \pm 0.25$ |
| *Recent methods* | | | | |
| ToolRL | $47.78 \pm 0.13$ | $66.27 \pm 0.57$ | $66.57 \pm 0.14$ | $\mathbf{6.95} \pm 0.52$ |
| LUFFY | $48.81 \pm 0.18$ | $75.68 \pm 0.30$ | $64.80 \pm 0.29$ | $4.62 \pm 0.44$ |
| GKD | $47.71 \pm 0.20$ | $67.50 \pm 0.74$ | $\underline{67.89} \pm 0.17$ | $3.95 \pm 0.42$ |
| *Our methods* | | | | |
| CKD | $49.61 \pm 0.31$ | $75.44 \pm 0.38$ | $66.09 \pm 0.20$ | $5.12 \pm 0.54$ |
| Sim-RL | $49.56 \pm 0.15$ | $75.51 \pm 0.32$ | $67.52 \pm 0.20$ | $3.33 \pm 0.11$ |
| SFT+Sim-RL[*] | $\underline{50.30} \pm 0.14$ | $\underline{76.78} \pm 0.35$ | $66.81 \pm 0.20$ | $5.62 \pm 0.30$ |
| CKD+Sim-RL | $\mathbf{51.80} \pm 0.12$ | $\mathbf{78.98} \pm 0.29$ | $\mathbf{68.41} \pm 0.30$ | $\underline{6.93} \pm 0.48$ |

[†]Results from the official benchmark; error bars not provided.
[*]It refers to Sim-RL on SFT-think.

Table 8: Performance comparison of different fine-tuning methods on Qwen3-0.6B, evaluated on the ACEBench Normal benchmark. Values are presented as mean $\pm$ std.

| Method | Summary | Atom | Single-Turn | Multi-Turn | Similar API | Preference |
|---|---|---|---|---|---|---|
| *Standard methods* | | | | | | |
| Base-model[†] | 27.20 | 37.70 | 19.50 | 10.00 | 36.00 | 6.00 |
| SFT | $2.03 \pm 0.16$ | $1.23 \pm 0.41$ | $0.16 \pm 0.23$ | $0.00 \pm 0.00$ | $16.67 \pm 1.88$ | $0.00 \pm 0.00$ |
| SFT-think | $30.17 \pm 1.28$ | $44.47 \pm 1.67$ | $14.62 \pm 1.63$ | $12.75 \pm 2.16$ | $29.00 \pm 4.12$ | $11.49 \pm 2.59$ |
| FKL | $37.76 \pm 1.52$ | $53.55 \pm 3.03$ | $19.43 \pm 2.06$ | $18.31 \pm 3.03$ | $42.37 \pm 4.42$ | $14.00 \pm 3.84$ |
| *Recent methods* | | | | | | |
| ToolRL | $27.77 \pm 1.00$ | $41.00 \pm 2.35$ | $12.25 \pm 1.03$ | $10.00 \pm 0.70$ | $35.50 \pm 3.84$ | $7.50 \pm 2.17$ |
| LUFFY | $\underline{44.83} \pm 0.97$ | $\underline{60.56} \pm 1.56$ | $25.50 \pm 2.48$ | $\underline{22.99} \pm 2.16$ | $\underline{56.00} \pm 4.32$ | $\underline{22.00} \pm 1.63$ |
| GKD | $37.36 \pm 1.17$ | $49.43 \pm 2.04$ | $21.50 \pm 1.08$ | $18.66 \pm 0.94$ | $48.00 \pm 4.32$ | $\mathbf{23.33} \pm 4.98$ |
| *Our methods* | | | | | | |
| CKD | $39.71 \pm 1.20$ | $55.94 \pm 2.09$ | $20.51 \pm 2.24$ | $20.00 \pm 2.77$ | $43.06 \pm 4.87$ | $16.87 \pm 4.12$ |
| Sim-RL | $40.63 \pm 0.83$ | $54.76 \pm 1.11$ | $\underline{25.83} \pm 1.02$ | $\underline{22.99} \pm 4.08$ | $50.66 \pm 3.39$ | $10.66 \pm 2.49$ |
| SFT+Sim-RL[*] | $38.35 \pm 0.84$ | $52.80 \pm 1.43$ | $22.59 \pm 2.17$ | $19.56 \pm 3.11$ | $44.43 \pm 4.54$ | $14.43 \pm 3.59$ |
| CKD+Sim-RL | $\mathbf{52.11} \pm 0.91$ | $\mathbf{69.76} \pm 1.51$ | $\mathbf{33.67} \pm 1.62$ | $\mathbf{30.66} \pm 3.09$ | $\mathbf{57.67} \pm 4.10$ | $21.33 \pm 1.94$ |

[†]Results from the official benchmark; error bars not provided.
[*]It refers to Sim-RL on SFT-think.

Table 9: Model performance on function calling benchmarks with different KD strategies. Values are presented as mean $\pm$ std.

| Method | BFCLv3 Overall | | ACEBench Normal | |
| --- | --- | --- | --- | --- |
| | w/o RL | w/ RL | w/o RL | w/ RL |
| CE | $47.14 \pm 0.20$ | $50.36 \pm 0.10$ | $30.17 \pm 1.28$ | $38.35 \pm 0.84$ |
| FKL | $49.68 \pm 0.12$ | $\underline{51.16} \pm 0.14$ | $37.76 \pm 1.52$ | $49.96 \pm 1.04$ |
| RSKD | $49.34 \pm 0.29$ | $50.50 \pm 0.11$ | $38.10 \pm 0.92$ | $\underline{50.53} \pm 0.54$ |
| RKL[*] | $49.29 \pm 0.26$ | $50.58 \pm 0.06$ | $36.19 \pm 1.07$ | $40.56 \pm 0.37$ |
| AKL[*] | $49.59 \pm 0.39$ | $50.40 \pm 0.10$ | $43.80 \pm 1.23$ | $49.33 \pm 0.77$ |
| CKD | $49.61 \pm 0.31$ | $\mathbf{51.80} \pm 0.12$ | $39.71 \pm 1.20$ | $\mathbf{52.11} \pm 0.91$ |

[*]The stable variant.

Table 10: Ablation study on reward designs. Values are presented as mean $\pm$ std.

| Method | BFCLv3 Overall | ACEBench Normal |
| --- | --- | --- |
| CKD+Binary Reward | $50.83 \pm 0.09$ | $37.43 \pm 0.90$ |
| CKD+ToolRL | $49.01 \pm 0.13$ | $\underline{43.00} \pm 1.51$ |
| CKD+SwiRL | $\underline{50.96} \pm 0.09$ | $37.63 \pm 2.05$ |
| CKD+Sim-RL | $\mathbf{51.80} \pm 0.12$ | $\mathbf{52.11} \pm 0.91$ |

## A.11 CASE STUDY

This appendix provides a qualitative analysis illustrating how Sim-RL addresses critical failure modes present in other reward designs.

Table 11 demonstrates the rigidity of binary rewards. Functionally correct outputs—such as a function call missing an optional argument or containing a trivial formatting difference—are incorrectly assigned a score of 0. This provides no useful learning signal. Sim-RL resolves this by assigning partial credit for correct function and primary argument (0.5) and a full score for semantically equivalent outputs (1.0), thus rewarding genuine progress.

Table 12 highlights that even some well-performing RL methods can still encounter the issue of "reward hacking." For instance, a tool call may receive a perfect score (1.0) because it is syntactically correct for the user's immediate question. However, by ignoring the conversation history, this rewards an inefficient and redundant action if the model already has the answer. The model then learns to game the system by making simple, unnecessary tool calls, exacerbating this inefficient behavior. Sim-RL avoids this by comparing the action against the optimal context-aware response (a direct answer) and correctly assigns a score of 0.0 to penalize the suboptimal action.

In summary, Sim-RL combines semantic flexibility to handle near-correctness with contextual grounding to prevent reward hacking, resulting in a more robust and reliable reward signal for training agents.

Table 11: Binary Reward vs. Sim-RL for Partially Correct Tool Calls.

| | Example 1: Missing a Default Argument | Example 2: Trivial Formatting Difference |
|---|---|---|
| **Function** | `check_wordpress` | `label_template_brands` |
| **Query** | "Can you check if `https://example.com` is running WordPress?" | "Can you list the brands available for A4 size blank label sheets?" |
| **Model Rollout** | `{"name": "check_wordpress", "arguments": {"url": "https://example.com"}}` | `{"name": "label_template_brands", "arguments": {"format": "a4"}}` |
| **Ground Truth** | `{"name": "check_wordpress", "arguments": {"url": "https://example.com", "user_agent": "Mozilla/5.0"}}` | `{"name": "label_template_brands", "arguments": {"format": "A4"}}` |
| **Binary RL Score** | **0** (Mismatch) | **0** (Mismatch) |
| **Sim-RL Score** | **0.5** (Partial credit for correct function and primary argument) | **1.0** (ROUGE-L is case-insensitive) |

Table 12: Example of Reward Hacking via a Redundant Tool Call

| | **Example: Redundant Tool Call (Reward Hacking)** |
|---|---|
| **Context** | In a previous turn, the model already looked up information for "SFO" airport. |
| **Query** | "What is the ICAO code for SFO airport, and how many runways does it have?" |
| **Model Rollout** | `<tool_call> {"name": "airportstatistics", "arguments": {"iata": "SFO"}} </tool_call>` |
| **Ground Truth** | "The ICAO code for SFO is KSFO, and it has 4 runways." |
| **SwiRL Score** | **1.0** (Rewards the valid-looking tool call, ignoring context) |
| **Sim-RL Score** | **0.0** (Penalizes the unnecessary call compared to the optimal response) |

## A.12 ADDITIONAL RESULTS

To strengthen our claims and provide deeper insights, we have conducted the additional experiments and incorporated a more detailed analysis.

**Comparison on a Larger Student Model:** We apply our method to a 1.7B student model and compare its performance against the SFT+Sim-RL baseline. As Table 13 shows, CKD continues to outperform SFT, confirming the scalability and effectiveness of our approach on larger models.

**Ablation on Teacher Model Size:** We also conduct an ablation study on the teacher model's size, using a Qwen3-14B model as the teacher. The results in Table 14 below show that our method remains effective, demonstrating its robustness to the choice of teacher model size.

**Ablation on Teacher Refinement:** We run an ablation study comparing Qwen3-0.6B students distilled from the base teacher vs. the refined teacher, which is shown in Table 15. The results show that while a better teacher indeed leads to a better student, this does not affect the overall validity of our method. With the un-refined teacher, our core STAR method (CKD + Sim-RL) still clearly outperforms the standard SFT+Sim-RL baseline. Furthermore, without refinement, the suboptimal base teacher model leads to the comparatively lower performance of the student model after only the CKD stage. However, the subsequent Sim-RL to this student model results in a substantial performance gain. These observations are crucial as they substantiate the robustness of our framework, demonstrating its effectiveness even when initialized with a less capable teacher model.

Table 13: Performance comparison of CKD and SFT, followed by Sim-RL. The teacher model is the refined Qwen3-8B, and the student model is Qwen3-1.7B.

| Method | BFCLv3 Overall | ACEBench Normal |
|---|---|---|
| SFT to Qwen3-1.7B + Sim-RL | 55.54 | 56.20 |
| CKD to Qwen3-1.7B + Sim-RL | **56.05** | **60.90** |

Table 14: Ablation study on teacher model size. The student model is Qwen3-0.6B.

| Method | BFCLv3 Overall | | ACEBench Normal | |
|---|---|---|---|---|
| | w/o RL | w/ RL | w/o RL | w/ RL |
| CKD from Qwen3-8B + Sim-RL | 49.56 | 51.70 | 39.00 | 53.00 |
| CKD from Qwen3-14B + Sim-RL | 50.12 | 50.75 | 38.50 | 54.30 |

Table 15: Ablation study on teacher refinement. The teacher model is Qwen3-8B and the student model is Qwen3-0.6B.

| Method | BFCLv3 Overall | | ACEBench Normal | |
|---|---|---|---|---|
| | w/o RL | w/ RL | w/o RL | w/ RL |
| SFT+Sim-RL | 47.59 | 50.41 | 28.70 | 38.90 |
| CKD from Qwen3-8B (Base)+Sim-RL | 47.13 | 51.35 | 31.40 | 47.20 |
| CKD from Qwen3-8B (Refined)+Sim-RL | 49.56 | 51.70 | 39.00 | 53.00 |

