# OpenReview forum: "STAR: Similarity-guided Teacher-Assisted Refinement for Super-Tiny Function Calling Models"
_ICLR.cc/2026/Conference — ICLR 2026 Poster_

### Official Review · Reviewer_Qjs5 · 2025-10-27

**Soundness:** 1
**Presentation:** 2
**Contribution:** 2
**Rating:** 2
**Confidence:** 4

**Summary:**

This paper explores training extremely small language models for function calling. The authors introduce the STAR framework, which extends the traditional Knowledge Distillation (KD) + Reinforcement Learning (RL) pipeline with two innovations. First, Constrained Knowledge Distillation (CKD) mitigates key shortcomings of existing KD methods. Second, during the RL phase, SimRL employs a similarity-based reward function aligned with ground-truth outputs. Through evaluations on standard benchmarks, the authors demonstrate that STAR models surpass state-of-the-art KD and RL approaches within the super-tiny model regime.

**Strengths:**

Strengths:

1. The paper is well written and easy to follow, the experimental results are also well presented.

2. The CKD loss is well motivated, and the gradient analysis is particularly insightful.

3. The final performance of the STAR models are impressive.

4. The paper shows that current SOTA  RL / distillation methods struggle in the super tiny models regime, which is interesting.

**Weaknesses:**

The two main contributions of this paper are 1) the CKD loss function and 2) the sim-rl reward function, and the impact of both has not been well studied.

1. In Table 4, CKD’s performance does not appear significantly stronger than other loss functions. For instance, CKD outperforms AKL by only a small margin (~0.1–0.5) on the BFCLv3 benchmark but performs worse than AKL on ACEBench without RL. Were the improvements of CKD tested for statistical significance?

2. The proposed reward function has not been evaluated against prior approaches—for instance, the similarity-based reward function in ToolRL or other variants based on ASTs and PRMs. As a result, its effectiveness remains unvalidated.

3. There is no ablation on the effect of distilling from a teacher; what if STAR is applied to D directly?

4. (Line 418) “standard metrics are unreliable..” – what are the standard metrics? I also didn’t follow how sim-rl is better suited to handling the stochasticity of the teacher?

5. (Minor) Since CKD is applied first in the pipeline, the paper reads better if it’s introduced first.

6. (Minor) A lot of citations are missing a space after the text, for ex. Line 38: “function calling(Patil et al., 2024; Jin et al., 2025)”

**Questions:**

1. Is the improvement of CWD on baselines (such as AKL) statistically significant?

2. How does the Sim-RL reward function compare to other reward functions in the literature (ex: ToolRL)?

3. What is the impact of using the teacher’s generations D^T v/s D?

4. In KD, the asymmetry of the divergence seems to be leading to poor performance. What if you replace it with a symmetric divergence? Ex: Jensen Shannon

---

> ### Author Response · Authors · 2025-11-21
> **Response to Reviewer Qjs5 [1/4]**
>
> We sincerely thank you for the detailed and constructive feedback. We appreciate the acknowledgment of our paper's writing, presentation, and the impressive performance of our STAR models. Your critiques have been invaluable in helping us strengthen our work. We have performed substantial revisions, including new experiments and in-depth analyses, to address every concern raised. We believe these changes can help to improve the soundness and contribution of our paper.
>
> Below, we provide a point-by-point response to the weaknesses and questions.
>
> ------
>
> ### **Replies to Weaknesses and Questions**
>
> **1. Regarding the impact and statistical significance of CKD (Weakness 1, Question 1)**
>
> >*"In Table 4, CKD’s performance does not appear significantly stronger than other loss functions... Were the improvements of CKD tested for statistical significance?" and "Is the improvement of CWD on baselines (such as AKL) statistically significant?"*
>
> **Our Reply:** Thank you for this excellent suggestion. We have to clarify that the results in Table 4 are only a part of KD ablation experiments, which do not capture the full picture of CKD's advantages and statistical significance within our complete methodology. We apologize for not sufficiently making this clearer. To address this, we have made a more detailed clarification of the analysis in **Section 4.3** and added new visualizations (**Figure 3 and Figure 4 in the revised paper**) to illustrate the superiority of CKD compared to other methods, its statistical significance, its synergy with the subsequent Sim-RL phase, and its overall contribution to our STAR framework, supported by established literature.
>
> Our detailed analysis, supported by two well-established literature [1, 2], is as follows:
>
> - **The Trade-off of Mode-Seeking in Initial Performance:** It is an anticipated outcome that distillation methods incorporating Reverse KL-divergence (RKL), such as AKL, may exhibit a higher initial `Pass@1` score. As discussed in Section 3.1.2, the mode-seeking nature of RKL largely optimizes the student to replicate the teacher's single most probable output. While this can artificially inflate metrics like `Pass@1` before RL (as observed on ACEBench in Table 4), it comes at the cost of a sharply reduced policy entropy. This suppression of the probability distribution's tail limits the model's exploratory capacity, a crucial prerequisite for effective RL. Therefore, as argued in recent work [1], this narrow focus on the primary mode, `Pass@1`, is often a misleading indicator of a model's true reasoning capabilities and its potential for improvement with RL.
> - **Superior Reasoning Potential (Higher Pass@k):** As recent work has shown [1], a more robust predictor of a model's reasoning capabilities and its potential for improvement with RL is the `Pass@k` accuracy at larger k values, which reflects the model's ability to generate a diverse set of correct solutions. As shown in our new **Figures 3 and 4**, the CKD-initialized model achieves the highest `Pass@k` scores among all initialization methods, including SFT and other KD variants. This indicates that CKD endows the model with superior reasoning potential before RL even begins. This advantage stems from CKD's unique re-balancing of the learning signal: it preserves the teacher's top-k probabilities while introducing a targeted suppression term that forcefully penalizes "confident-but-wrong" logits, rather than aggressively collapsing the tail of the distribution.
> - **Enhanced Exploration Capacity (Higher Entropy):** A key requirement for successful RL is the policy's ability to explore the solution space. A higher policy entropy before RL training promotes effective exploration and prevents premature convergence to suboptimal solutions [2]. As shown in our new **Figure 4**, CKD-initialized policies exhibit significantly higher entropy at the start of RL training compared to other KD methods. This is because our focused distillation approach avoids the aggressive mode-seeking behavior of RKL-based methods, which tends to suppress entropy. This higher entropy facilitates more effective exploration during the Sim-RL phase, leading to better final performance.
>
> In essence, CKD provides a better starting point for Sim-RL, setting a higher ceiling for the model's final performance with both stronger reasoning and greater exploratory potential. This makes CKD's role critically important, as it largely determines the upper bound of our method's final performance. CKD synergizes most effectively with Sim-RL because it achieves higher `Pass@k` and policy entropy than other advanced methods, which are limited by a suppressed policy entropy. This new analysis provides the statistical and mechanistic evidence for CKD's significance that was not fully presented before.

---

> ### Author Response · Authors · 2025-11-21
> **Response to Reviewer Qjs5 [2/4]**
>
> **2. Regarding the comparison of Sim-RL with other reward functions (Weakness 2, Question 2)**
>
> >*"The proposed reward function has not been evaluated against prior approaches—for instance, the similarity-based reward function in ToolRL or other variants based on ASTs and PRMs. As a result, its effectiveness remains unvalidated." and "How does the Sim-RL reward function compare to other reward functions in the literature (ex: ToolRL)?"*
>
> **Our Reply:** Thank you for this valuable suggestion. To validate the effectiveness of our Sim-RL reward function, we have conducted new ablation studies comparing it against several prior reward functions. We compared Sim-RL with **Binary Reward** [3], the reward from **ToolRL** [4], and the **SwiRL,** a PRM-based approach [5]. Please note that for SwiRL, which uses a Process Reward Model (PRM), the original work employed Gemini-1.5-Pro. Due to cost constraints, we used Qwen3-8B as the PRM in our experiment. All experiments were conducted on the Qwen3-0.6B model initialized with CKD, with only the reward function being varied.
>
> The results, now presented in **Table 5** in the revised paper, are as follows:
>
> | Method                | BFCLv3 Overall | ACEBench Normal |
> | --------------------- | -------------- | --------------- |
> | CKD+Binary Reward     | 51.05          | 35.70           |
> | CKD+ToolRL            | 48.59          | 40.50           |
> | CKD+SwiRL             | 51.10          | 40.30           |
> | **CKD+Sim-RL (Ours)** | **51.70**      | **53.00**       |
>
> The results clearly demonstrate that Sim-RL provides a significant advantage over the other methods, especially on the more challenging ACEBench generalization benchmark. Our analysis is as follows:
>
> - **Binary Reward** is too strict. It only rewards rollouts that perfectly match the ground truth, leading to high precision but low recall. This approach struggles in multi-solution scenarios and fails to generalize well, as seen in its poor ACEBench score.
> - **ToolRL** and **Sim-RL** both use a dense reward, but our **Sim-RL** provides a more comprehensive signal by also rewarding non-call text based on similarity, which enhances its generalization capabilities.
> - **SwiRL (The PRM-based approach)** with a Qwen3-8B PRM underperforms Sim-RL. While a more powerful PRM could potentially yield better results, using PRMs introduces significant training costs and the risk of "reward hacking" from a black-box model [6]. Sim-RL, in contrast, is low-cost, interpretable, and empirically more effective in our setup.
>
> This new analysis and the corresponding results have been integrated into **Section 4.3** to better showcase the effectiveness of Sim-RL.

---

> ### Author Response · Authors · 2025-11-21
> **Response to Reviewer Qjs5 [3/4]**
>
> **3. Regarding the impact of using teacher-generated data (D^T) vs. ground-truth data (D) (Weakness 3, Question 3)**
>
> >*"There is no ablation on the effect of distilling from a teacher; what if STAR is applied to D directly?" and "What is the impact of using the teacher’s generations D^T v/s D?"*
>
> **Our Reply:** Thank you for this insightful question. First, we apologize for the lack of clarity in our original Tables 1 and 2. The entry "SFT+Sim-RL" was indeed trained on the teacher-augmented dataset $\mathcal{D}_T$ (i.e., it is equivalent to SFT-think followed by Sim-RL). Both "SFT+Sim-RL" and "CKD+Sim-RL" were trained on the same teacher-augmented dataset $\mathcal{D}_T$. We have clarified this in the captions of **Tables 1 and 2** in the revised paper.
>
> Per your recommendation, we conducted a new experiment applying CKD directly on the ground-truth data $\mathcal{D}$ (which lacks Chain-of-Thought, CoT). The results are shown in the table below and will be added to the appendix of our revised paper at the end of the rebuttal.
>
> Performance comparison of CKD on different training datasets. Scores are reported on the BFCL and AceBench benchmarks. The student model (Qwen3-0.6B) and the teacher model (Qwen3-8B) are trained via CKD:
>
> |                | BFCLv3 Overall | AceBench Normal |
> | -------------- | -------------- | --------------- |
> | CKD on $\mathcal{D}$       | 35.77          | 23.70           |
> | **CKD on $\mathcal{D}_T$** | **49.56**      | **39.00**       |
>
> We found that distilling on $\mathcal{D}$ yielded very poor performance, even underperforming the SFT-think baseline. We attribute this to a **fundamental distributional mismatch**:
>
> - The teacher's output distribution is conditioned on generating a CoT reasoning process.
> - The ground-truth data $\mathcal{D}$ represents a direct-answer distribution without reasoning steps.
>
> Distilling from a CoT-generating teacher onto a direct-answer target forces the student to learn a mismatched mapping and discards the most valuable signal—the soft labels over the reasoning steps. This prevents the student from learning the teacher's reasoning capabilities. Our approach of distilling on complete teacher-generated rollouts $\mathcal{D}_T$ (including CoT) is specifically designed to transfer this crucial reasoning process. This is a conventional and validated technique for teaching reasoning, as evidenced by its use in training state-of-the-art models like Qwen3 [7] and DeepSeek-R1 [8], and is essential for achieving the reasoning improvements our paper presents.
>
> ------
>
> **4. Regarding the clarity of Sim-RL's benefits and "standard metrics" (Weakness 4)**
>
> >(Line 418) "standard metrics are unreliable.." – what are the standard metrics? I also didn’t follow how sim-rl is better suited to handling the stochasticity of the teacher?
>
> **Our Reply:** We apologize for the lack of clarity in that sentence. The "standard metrics" we referred to are the **binary reward** [3], which assigns a reward of 1 for a perfect match and 0 otherwise.
>
> Regarding how Sim-RL better handles stochasticity: a binary reward is brittle because it harshly penalizes functionally correct yet syntactically varied solutions, which are common when dealing with teacher models that can generate multiple valid answers for the same prompt. Our Sim-RL is more robust because its fine-grained, continuous reward signal evaluates the functional **similarity** of the output rather than enforcing a strict pass/fail criterion. This richer signal more effectively guides the policy towards the diverse set of valid solutions, enhancing generalization and confirming that a task-aligned similarity metric is crucial for optimal policy refinement.
>
> We have made a clearer clarification of this point and added the ablation study in **Table 5** of **Section 4.3** in the revised paper. Also, the reply to Weakness 2 and Question 2 can help explain this.

---

> ### Author Response · Authors · 2025-11-21
> **Response to Reviewer Qjs5 [4/4]**
>
> **5. Regarding the suggestion to use a symmetric divergence like Jensen-Shannon (Question 4)**
>
> >*"In KD, the asymmetry of the divergence seems to be leading to poor performance. What if you replace it with a symmetric divergence? Ex: Jensen Shannon"*
>
> **Our Reply:** Thank you for this excellent technical suggestion. Following your advice, we conducted new experiments replacing the FKL component of our loss with the symmetric Jensen-Shannon Divergence (JSD). We tested both a standard top-k JSD and a stabilized variant (JSD*) that includes our tail suppression term. The results are presented below:
>
> | Method  | BFCLv3 Overall | |  ACEBench Normal           |           |
> | ------- | -------------- | --------------- | --------- | --------- |
> | |w/o RL  | w/ RL          | w/o RL          | w/ RL     |
> | JSD     | 48.62          | 50.91           | 35.80     | 40.80     |
> | JSD*    | 48.09          | 50.69           | 42.70     | 44.90     |
> | **CKD** | **49.56**      | **51.70**       | **39.00** | **53.00** |
>
> \*The stable variant. We perform KD using top-k JSD with and without tail suppression term, denoted as JSD and JSD*
>
>
>
> As the results show, our CKD outperforms both JSD-based variants. While JSD is symmetric, it still exhibits mode-seeking tendencies compared to the distribution-covering nature of CKD. In the specific context of training super-tiny models for function calling, maintaining a broader probability distribution (high entropy) is crucial for the student to retain the capacity for exploration during the RL stage. CKD successfully forces the student to cover the teacher's distribution, whereas JSD's compromise between modes can prematurely narrow the policy, limiting the effectiveness of the subsequent refinement.
>
>
>
> ------
>
> **6. Regarding Minor Presentation Issues (Weaknesses 5 & 6)**
>
> >*"(Minor) Since CKD is applied first in the pipeline, the paper reads better if it’s introduced first. (Minor) A lot of citations are missing a space after the text."*
>
> **Our Reply:** Thank you for these helpful presentation suggestions.
>
> - We agree completely. In our revised manuscript, we have reordered **Section 3** to introduce **CKD (now Section 3.1)** before **Sim-RL (now Section 3.2)**, which aligns with the chronological flow of our framework.
> - We apologize for the formatting errors. We have carefully reviewed the entire manuscript and corrected all instances of missing spaces before citations.
>
> ------
>
> We are grateful for your thorough and insightful comments, which have pushed us to significantly strengthen our paper. Through new experiments, deeper analysis, and clearer presentation, we have provided substantial new evidence for the effectiveness and significance of both CKD and Sim-RL.
>
> We hope our response has clearly addressed your concerns, and we welcome further discussion. If you recognize the value of our method and its improved presentation, we would be deeply grateful if you would consider raising your score. Thank you very much.
>
>
>
>
>
> **References:**
>
> [1] Kang et al. Quagmires in sft-rl post-training: When high sft scores mislead and what to use instead, 2025. URL https://arxiv.org/abs/2510.01624.
>
> [2] Cui et al. The entropy mechanism of reinforcement learning for reasoning language models, 2025b. URL https://arxiv.org/abs/2505.22617.
>
> [3] Hao et al. Exploring Superior Function Calls via Reinforcement Learning, 2025. URL https://arxiv.org/abs/2508.05118v3.
>
> [4] Qian et al. Toolrl: Reward is all tool learning needs, 2025. URL https://arxiv.org/abs/2504.13958.
>
> [5] Goldie et al. Synthetic data generation and multi-step reinforcement learning for reasoning and tool use. In Second Conference on Language Modeling, 2025. URL https://openreview.net/forum?id=oN9STRYQVa.
>
> [6] Khalaf et al. Inference-Time Reward Hacking in Large Language Models, 2025. URL https://arxiv.org/abs/2506.19248.
>
> [7] Yang et al. Qwen3 technical report, 2025. URL https://arxiv.org/abs/2505.09388.
>
> [8] Guo et al. Deepseek-r1: Incentivizing reasoning capability in llms via reinforcement learning, 2025. URL https://arxiv.org/abs/2501.12948

---

> > ### Comment · Reviewer_Qjs5 · 2025-11-26
> >
> > Thank you for the detailed responses; my concerns regarding Questions 2 and 3 have been addressed. However, it is still not evident to me that the reported CKD improvements are statistically significant.
> >
> > 1. Lack of variance reporting: Why are there no error bars in the figures and tables? How many random seeds were used for each experiment?
> >
> > 2. Small improvements: As mentioned previously, the differences reported in Table 4 are very small—often less than 1 percentage point. Without a robust statistical analysis and results aggregated over multiple seeds, the effectiveness of CKD remains ambiguous.
> >
> > 3. Figure 4: The entropy-based analysis is interesting, but in Figure 4, the performance curves for CKD and AKL appear nearly identical until k=26. Given how close the curves are, it is difficult to conclude that CKD provides a meaningful improvement or that a higher entropy leads to greater pass@k performance in this task.
> >
> > 4. From Figure~3, the impact of Sim\text{-}RL appears quite limited, as its performance lags behind CKD for approximately $k > 12$.

---

> ### Author Response · Authors · 2025-11-27
> **Response to Reviewer Qjs5 [1/2]**
>
> Thank you for your continued engagement and for acknowledging that your concerns regarding Questions 2 and 3 have been addressed. However, we are concerned that the current assessment, which questions the statistical significance of our Constrained Knowledge Distillation (CKD), seems to **focus on the isolated, incremental performance of this single component and overlooks the central thesis of** **our paper**. Our central contribution is **the holistic STAR framework,** which demonstrates a novel and effective integration of CKD and Sim-RL to achieve state-of-the-art results for super-tiny models. The value of our components, especially CKD, **cannot be judged in isolation but must be evaluated by their direct and substantial contribution to the final performance of the complete framework.** We wish to clarify our main contributions and the robust, statistically significant advantages our method provides. To address your new concerns, we have now also provided robust statistical analysis and further clarifications below.
>
> ------
>
> ### **Regarding the Overall Contribution and Statistical Significance of STAR (Points 1 & 2)**
>
> - **Holistic Framework Superiority:** We must first re-emphasize the primary goal and contribution of our paper: to demonstrate that the complete STAR framework significantly advances the state-of-the-art for super-tiny function-calling models. As shown in our original Tables 1 and 2, our STAR-0.6B model achieves **51.70 on BFCLv3 and 53.00 on ACEBench**, establishing a new SOTA in the 0.6B model class. **This performance decisively outperforms the mainstream and widely-used methods like SFT-think (47.59 / 28.70) and conventional SFT+RL approaches like ToolRL (47.35 / 29.40).** This large performance gap is the core result of our work and underscores the effectiveness of our combined CKD and Sim-RL approach.
>
>   | Method               | BFCLv3 Overall | ACEBench Normal |
>   | -------------------- | -------------- | --------------- |
>   | Base Model           | 47.33          | 27.20           |
>   | SFT-think            | 47.59          | 28.70           |
>   | ToolRL               | 47.35          | 29.40           |
>   | **STAR (Ours)** | **51.70**      | **53.00**       |
>
> - **Statistical Significance and High Stability:** To address your concerns about variance and significance (Points 1 and 2), we have conducted experiments using **3 different random seeds**. The results, including mean and standard deviation, are presented in the table below. This new data provides robust statistical evidence for CKD's superiority over the strong Stabilized AKL baseline, not just in its final score, but more importantly, in its potential to benefit from RL.
>
>   | **Method** |       BFCLv3 Overall | | ACEBench Normal |                            |
>   | ------------- | -------------- | --------------- | ------------ | ------------ |
>   |  | w/o RL        | w/ RL          | w/o RL          | w/ RL        |
>   | CE            | 47.14 ± 0.20   | 50.36 ± 0.10    | 30.17 ± 1.28 | 38.35 ± 0.84 |
>   | FKL           | 49.68 ± 0.12   | 51.16 ± 0.14    | 37.76 ± 1.52 | 49.96 ± 1.04 |
>   | RSKD          | 49.34 ± 0.29   | 50.50 ± 0.11    | 38.10 ± 0.92 | 50.53 ± 0.54 |
>   | Stablized RKL | 49.29 ± 0.26   | 50.58 ± 0.06    | 36.19 ± 1.07 | 40.56 ± 0.37 |
>   | Stablized AKL | 49.59 ± 0.39   | 50.40 ± 0.10    | 43.80 ± 1.23 | 49.33+0.77   |
>   | CKD           | 49.61 ± 0.31   | **51.80 ± 0.12**    | 39.71 ± 1.20 | **52.11 ± 0.91** |
>
>   As the table demonstrates:
>
>   - **High Stability and Robustness:** The standard deviations across all methods are consistently low. Crucially, the final performance gap between CKD+Sim-RL and Stablized AKL+Sim-RL on both benchmarks is **more than 3 times larger than the standard deviation of either result**. This strongly indicates that the observed improvement is statistically significant and robust, not an artifact of random noise.
>   - **Not "Small Improvements", but Greater RL Potential:** CKD consistently enables far greater performance gains from the Sim-RL phase. On BFCLv3, CKD gains **+2.19** points (49.61 to 51.80) from its initial state, whereas AKL only gains **+0.81** (49.59 to 50.40). The difference is even more stark on the challenging ACEBench generalization benchmark, where CKD gains **+12.40** points (39.71 to 52.11) compared to AKL's **+5.53** (43.80 to 49.33). This shows that CKD creates a much better policy initialization for RL, which is a key part of our claim.

---

> ### Author Response · Authors · 2025-11-27
> **Response to Reviewer Qjs5 [2/2]**
>
> ### **Regarding the Interpretation of Pass@k and Entropy (Point 3)**
>
> - **Causality of Performance via Policy Entropy:** We agree that the `Pass@k` curves for CKD and AKL in Figure 4 appear close for larger values of *k*. However, the crucial takeaway is not the curve itself, but the **underlying cause** of the final performance difference after RL. As shown in **Figure 4 (top)**, the CKD-initialized policy maintains a **significantly and visibly higher policy entropy** than the AKL-initialized policy throughout training. This large gap in entropy is the critical mechanistic difference we aimed to highlight. As established in the literature [1], a higher-entropy policy possesses a greater exploratory capacity, which is an essential precondition for successful and efficient RL refinement. We must also clarify that **we do not claim that higher entropy directly leads to higher** `Pass@k` **scores.** The precise relationship between policy entropy and `Pass@k` is a complex and interesting research question, but it is beyond the scope of this work. We believe this represents a promising direction for future investigation.
>
> ### **Regarding the Role of Sim-RL in Optimization (Point 4)**
>
> - **Misinterpretation of RL's Optimization Goal:** The observation that the `CKD+Sim-RL` curve "lags behind" the `CKD` curve for k>12 in Figure 3 stems from a misunderstanding of the RL process. This behavior is not a limitation but is, in fact, proof that our Sim-RL phase is working exactly as intended.
>   - The **CKD** curve shows the policy **before** RL. Its high `Pass@k` score indicates that the distillation phase has successfully taught the model a *diverse set of plausible solutions*, giving it broad reasoning potential.
>   - The **CKD+Sim-RL** curve shows the final policy **after** RL. The fundamental goal of the RL phase is to maximize task success, which corresponds to maximizing `Pass@1`. To achieve this, the optimization process correctly learns to concentrate the policy's probability mass on the single best-predicted solution.
>   - **The "lag" you observed is not a limitation but evidence that the RL optimization is working as intended. It successfully converts the broad potential of the CKD policy into a specialized, high-accuracy policy at the cost of the `Pass@k` [2],** validated by the significant performance jump from "w/o RL" to "w/ RL" in our new statistical table.
>
> We believe our clarifications fully address the remaining concerns and demonstrate the robust, statistically significant contributions of our work.
>
>
>
> [1] Cui et al. The entropy mechanism of reinforcement learning for reasoning language models. arxiv 2505.22617.
>
> [2] Deng et al. From Trial-and-Error to Improvement: A Systematic Analysis of LLM Exploration Mechanisms in RLVR. arxiv 2508.07534.

---

### Official Review · Reviewer_u15k · 2025-11-01

**Soundness:** 3
**Presentation:** 3
**Contribution:** 2
**Rating:** 6
**Confidence:** 3

**Summary:**

The paper introduces a training pipeline to adapt tiny-scale language models for function calling. Specifically, the introduced pipeline is composed of a RL method, Sim-RL, with rewards crafted based on the similarity between generated function calls against the ground truth and a knowledge distillation method, Constrained Knowledge Distillation (CKD), that uses forward KL-divergence as the divergence metrics with an additional component to penalize the confident-but-wrong tokens. The combination of these two approaches, STAR, shows the best performance among models under 1B and closes the performance gap with larger models on two benchmarks.

**Strengths:**

- The paper is relatively well-written and easy to follow.
- The performance gains on the 0.6B model scale is consistent over most metrics in the two evaluation benchmarks, and the performance gap with larger is significantly reduced.
- The paper includes discussion on the comparison between KD+RL and SFT+RL besides the empirical results that might be insightful for future work.
- The paper includes analysis on the comparison among different KD methods.

**Weaknesses:**

- Sim-RL looks highly dependent on the generated function calls’ format that the author defines based on the Qwen tool calling template. It might be important to show the generalizability of this method for alternative formats.
- STAR requires RL training on the teacher model, which introduces additional non-trivial compute cost compared to SFT+RL.
- More potential analysis studies could improve the persuasiveness of the paper in showing the advantages of CKD over SFT. For example, a comparison between these methods in a larger model (e.g. 1.7B), an ablation on the teacher model’s size, etc.

**Questions:**

- It seems that the methodology of CKD is not specific to the task of generating function calls. Has the author considered applying this method to other tasks? If not, what constrains CKD to this specific task?
- The paper lacks some explanations on the categories of the benchmarks shown in Table 1 and 2.

---

> ### Author Response · Authors · 2025-11-21
> **Response to Reviewer u15k [1/3]**
>
> We sincerely thank you for the thorough and constructive feedback. We are encouraged that you found our paper well-written, our performance gains consistent, and our analysis insightful. We appreciate the detailed comments, which have been invaluable in helping us improve the quality and clarity of our work.
>
> Below, we address your concerns and questions point-by-point. We have also revised the manuscript accordingly, and we hope our responses and revisions adequately address the points raised.
>
> ### **Response to Weaknesses**
>
> **1. On the generalizability of Sim-RL (Weakness 1)**
>
> >*“Sim-RL looks highly dependent on the generated function calls’ format that the author defines based on the Qwen tool calling template. It might be important to show the generalizability of this method for alternative formats.”*
>
> We sincerely apologize for the lack of clarity in our initial description that led to this impression. We would like to clarify that, based on the essence and nature of Sim-RL design, **in fact, Sim-RL is not dependent on any specific template.**
>
> Our method's components are designed for generalizability:
>
> - For the **Format Reward** term of our reward, a score of 1 is awarded as long as the generated output conforms to a predefined structure that allows for successful parsing, regardless of the specific template (e.g., XML-based, JSON-based, etc.).
> - For the **Answer Reward**, our similarity metric can be computed as long as the function call's name and arguments, or the final response, can be extracted from the given format.
>
> Generally, templates only have a slight impact on the performance of the initial policies or not [1]. The Qwen Chat Template was mentioned only as a concrete example for our specific implementation, but our method is by no means restricted to it. To address this, we have **revised the descriptions in Section 3.2.1 of the manuscript** to better clarify the general-purpose nature of our approach. We hope these changes make the universal applicability of Sim-RL clearer.
>
>
> **2. On the computational cost of STAR (Weakness 2)**
>
> >*“STAR requires RL training on the teacher model, which introduces additional non-trivial compute cost compared to SFT+RL.”*
>
> We thank the reviewer for raising this important practical concern. We would like to offer three points of clarification:
>
> 1. **Why did we do it? Optional, Not Required**: Our goal was to distill from the good teacher possible. By refining the base Qwen3-8B with Sim-RL, we could adapt it to our specific task and create a stronger teacher. This step is optional, not a mandatory part of the framework, but it helps get better performance, as mentioned in Section 3.3. Given the open-source models we had access to, the limited quality of the available data, and the various methods at our disposal (like standard RL, Sim-RL, SFT), we found that the Qwen3-8B model after being refined with Sim-RL was the relatively good teacher we could produce. Therefore, we chose to use this refined teacher to carry out the subsequent work, aiming to transfer the better possible capabilities to our student model.
> 2. **Modest Computational Cost:** In practice, the computational overhead for this optional step is modest. The teacher model refinement required only **160 H20 GPU hours**, which we believe is a relatively small cost for creating a high-quality teacher for distillation.
> 3. **Does it affect our main point? Ablation on Teacher Refinement**: To check this, we ran an ablation study comparing Qwen3-0.6B students distilled from the base teacher vs. the refined teacher, which is shown below. The results show that while a better teacher indeed leads to a better student, this does not affect the overall validity of our method. With the un-refined teacher, our core STAR method (CKD + Sim-RL) still clearly outperformed the standard SFT+Sim-RL baseline. Furthermore, without refinement, the base teacher model is suboptimal. Consequently, the student model's performance after only the CKD stage is comparatively lower. However, the subsequent Sim-RL to this student model results in a substantial performance gain. These observations are crucial as they substantiate the robustness of our framework, demonstrating its effectiveness even when initialized with a less capable teacher model.
>
> Ablation study on teacher model refinement is shown below. We are ready to add this study to the appendix in the future to make this analysis clearer.
>
> | Method                             | BFCLv3 Overall |       | ACEBench Normal |       |
> | ---------------------------------- | -------------- | ----- | --------------- | ----- |
> |                                    | w/o RL         | w/ RL | w/o RL          | w/ RL |
> | SFT+Sim-RL                         | 47.59          | 50.41 | 28.70           | 38.90 |
> | CKD from Qwen3-8B (Base)+Sim-RL    | 47.13          | 51.35 | 31.40           | 47.20 |
> | CKD from Qwen3-8B (Refined)+Sim-RL | 49.56          | 51.70 | 39.00           | 53.00 |

---

> ### Author Response · Authors · 2025-11-21
> **Response to Reviewer u15k [2/3]**
>
> **3. On the need for more analysis of CKD vs. SFT (Weakness 3)**
>
> >*"More potential analysis studies could improve the persuasiveness of the paper in showing the advantages of CKD over SFT. For example, a comparison between these methods in a larger model (e.g. 1.7B), an ablation on the teacher model’s size, etc."*
>
> We are very grateful for these excellent suggestions. To strengthen our claims and provide deeper insights, we have conducted the suggested experiments and incorporated a more detailed analysis into the paper.
>
> 1. **Comparison on a Larger Student Model (1.7B):** We applied our method to a 1.7B student model and compared its performance against the SFT+Sim-RL baseline. As the new table shows (to be added in the appendix), CKD continues to outperform SFT, confirming the scalability and effectiveness of our approach on larger models.
>
>    Performance comparison of CKD and SFT, followed by Sim-RL. The teacher model is the refined Qwen3-8B, and the student model is Qwen3-1.7B:
>
>    | **Method**                 | **BFCLv3 Overall** | **AceBench Normal** |
>    | -------------------------- | ------------------ | ------------------- |
>    | SFT to Qwen3-1.7B + Sim-RL | 55.54              | 56.2                |
>    | CKD to Qwen3-1.7B + Sim-RL | **56.05**          | **60.9**            |
>
> 2. **Ablation on Teacher Model Size:** We also conducted an ablation study on the teacher model's size, using a Qwen3-14B model as the teacher. The results in the new table below (to be added in the appendix) show that our method remains effective, demonstrating its robustness to the choice of teacher model size.
>
>
>    Ablation study on teacher model size. The student model is Qwen3-0.6B:
>
>    | **Method**                  | **BFCLv3 Overall** |           | **AceBench Normal** |           |
>    | --------------------------- | ------------------ | --------- | ------------------- | --------- |
>    |                             | **w/o RL**         | **w/ RL** | **w/o RL**          | **w/ RL** |
>    | CKD from Qwen3-8B + Sim-RL  | 49.56              | 51.70     | 39.0                | 53.0      |
>    | CKD from Qwen3-14B + Sim-RL | 50.12              | 50.75     | 38.5                | 54.3      |
>
> 3. **Deeper Analysis of CKD's Advantages:** To better demonstrate the advantages of CKD, we have made a better clarification of the analysis in **Section 4.3** and added new visualizations (**Figure 3 and Figure 4 in the revised paper**) to illustrate the superiority of CKD compared to other methods, its synergy with the subsequent Sim-RL phase, and its overall contribution to our STAR framework.
>
>    Our detailed analysis, supported by two well-established literature [2, 3], is as follows:
>
>    - **Superior Reasoning Potential (Higher Pass@k):** As recent work has shown [2], traditional performance metrics like `Pass@1` can be misleading indicators of a model's true reasoning capabilities and its potential for improvement with RL. A more robust predictor is the `Pass@k` accuracy at larger k values, which reflects the model's ability to generate a diverse set of correct solutions. As shown in our new **Figures 3 and 4**, the CKD-initialized model achieves the highest `Pass@k` scores among all initialization methods, including SFT and other KD variants. This indicates that CKD endows the model with superior reasoning potential before RL even begins. This advantage stems from CKD's unique re-balancing of the learning signal: it preserves the teacher's top-k probabilities while introducing a targeted suppression term that forcefully penalizes "confident-but-wrong" logits, rather than aggressively collapsing the tail of the distribution.
>    - **Enhanced Exploration Capacity (Higher Entropy):** A key requirement for successful RL is the policy's ability to explore the solution space. A higher policy entropy before RL training promotes effective exploration and prevents premature convergence to suboptimal solutions [3]. As shown in our new **Figure 4**, CKD-initialized policies exhibit significantly higher entropy at the start of RL training compared to other KD methods. This is because our focused distillation approach avoids the aggressive mode-seeking behavior of RKL-based methods, which tends to suppress entropy. This higher entropy facilitates more effective exploration during the Sim-RL phase, leading to better final performance.
>
> In essence, CKD provides a better starting point for Sim-RL, setting a higher ceiling for the model's final performance with both stronger reasoning and greater exploratory potential. This makes CKD's role critically important, as it largely determines the upper bound of our method's final performance. CKD synergizes most effectively with Sim-RL, because it achieves higher `Pass@k` and policy entropy than other advanced methods, which are limited by a suppressed policy entropy. This new analysis provides the statistical and mechanistic evidence for CKD's significance that was not fully presented before.

---

> ### Author Response · Authors · 2025-11-21
> **Response to Reviewer u15k [3/3]**
>
> ------
>
>
>
> ### **Response to Questions**
>
> **1. On the generalizability of CKD to other tasks (Question 1)**
>
> >*“It seems that the methodology of CKD is not specific to the task of generating function calls. Has the author considered applying this method to other tasks? If not, what constrains CKD to this specific task?”*
>
> This is an excellent observation, and we fully agree with your intuition. From a theoretical standpoint, CKD indeed holds strong potential for generalization as it introduces no domain-specific biases.
>
> We have considered its application to other domains and have even observed promising preliminary results and similar conclusions on the reasoning-gym benchmark, which supports its generalizability. However, for this paper, we chose to focus our scope on function calling to provide a deep and thorough investigation. Due to time and resource constraints, we have not yet prepared a complete set of experiments on other tasks that would meet the high standards of this conference.
>
> We strongly believe this is a promising direction for future research and plan to continue investigating the transferability of CKD to a wider range of reasoning tasks and agentic systems. We welcome further exploration in this area by the research community.
>
>
>
> **2. On the lack of explanation for benchmark categories (Question 2)**
>
> >*“The paper lacks some explanations on the categories of the benchmarks shown in Table 1 and 2.”*
>
> Thank you for pointing out this omission. We apologize for the lack of clarity. To address this, we have **added a new section, Appendix A.9,** which provides detailed explanations for all evaluation categories in the BFCLv3 and ACEBench benchmarks. We have also updated the captions of Tables 1 and 2.
>
> For convenience, here is a brief summary of the metric definitions:
>
> - **BFCLv3:**
>   - `Overall Acc`: Weighted average accuracy across all categories.
>   - `Non-Live Acc`: Accuracy on the static V1 dataset (prone to contamination).
>   - `Live Acc`: Accuracy on the unseen V2 "live" dataset to test generalization.
>   - `Multi-Turn Acc`: Accuracy on multi-turn and multi-step tasks.
> - **ACEBench-Normal:**
>   - `Summary`: Overall score for the Normal dataset.
>   - `Atom`: Performance on handling various parameter data types.
>   - `Single-Turn`: Accuracy in single-turn tool-calling scenarios.
>   - `Multi-Turn`: Accuracy in multi-turn dialogues, testing context management.
>   - `Similar API`: Ability to distinguish between nearly identical APIs.
>   - `Preference`: Ability to select APIs based on user preferences.
>
> ------
>
> We hope our responses and the corresponding revisions have clearly addressed your concerns. We are very grateful for the insightful feedback, which has significantly improved our paper.
>
> Thank you once again for your time and support. If you feel these revisions further improve the paper's contribution, we would be absolutely thrilled. We look forward to any further constructive feedback and positive assessment.
>
>
>
> **References:**
>
> [1] Liu et al. Understanding r1-zero-like training: A critical perspective, 2025b. URL https:[//arxiv.org/abs/2503.20783](http://arxiv.org/abs/2503.20783).
>
> [2] Kang et al. Quagmires in sft-rl post-training: When high sft scores mislead and what to use instead, 2025. URL https://arxiv.org/abs/2510.01624.
>
> [3] Cui et al. The entropy mechanism of reinforcement learning for reasoning language models, 2025b. URL https://arxiv.org/abs/2505.22617.

---

### Official Review · Reviewer_949b · 2025-11-03

**Soundness:** 3
**Presentation:** 3
**Contribution:** 3
**Rating:** 8
**Confidence:** 4

**Summary:**

This paper proposed a framework to effectively imbue tiny-LMs with tool calling capabilities of their larger models. The proposed method consists of two stages: (1) Constrained Knowledge Distillation to prevent highly confident incorrect predictions by the student model, and (2) a similarity-based RL refinement on top of the trained student that computes a fine-grained similarity-based reward between the ground-truth functional calls and the model prediction (more suitable for multi solutions problems). The authors verify their approach by comparing it to several baselines evaluated on BFCL and Acebench (for testing generalization). Results showed improved performance across all benchmarks with the more profound boost on Acebench showing better generalization to unseen function call formats.

**Strengths:**

- The paper is generally well-written and well motivated
- Experimental results are promising and show strong generalization compared to baselines
- The method seems to simple and effective at mitigating overfitting problem especially when compared to conventional approach of SFT+RL
- Results on closing the performance gap with much stronger models in Table 3 is pretty interesting
- Also appreciate additional theoretical explanation for their Top-K truncation with FK

**Weaknesses:**

- There’s no direct comparison with existing RL-based methods. It’s not clear how the proposed reward is different from those proposed in related prior works for example one in [1]. In general more comparison with existing RL rewards would be nice. (see more in questions)

[1] Anna Goldie, Azalia Mirhoseini, Hao Zhou, Irene Cai, and Christopher D Manning. Synthetic
data generation and multi-step reinforcement learning for reasoning and tool use.

**Questions:**

1- Line 057: typo: the -> the

2- Minor suggestion for structuring Sec 3: chronologically, it would have made sense to start with distillation and then talk about refinement (sim-RL)

3- [line 106] Add citation for RLVR: Lambert, Nathan, et al. 2025. “Tulu 3: Pushing Frontiers in Open Language Model Post-Training.” In Proceedings of the Second Conference on Language Modeling

4- [Line 270]: If I understand correctly, you have two iterations of Sim-RL? you refine both the teacher and the student using Sim-RL.  It would make sense to try to clarify this both in text and figure to avoid confusion.

5- Baslines: which baseline is representing a simple binary RL reward? This is specially important and relevant to you analysis section explaining your reward design

---

> ### Author Response · Authors · 2025-11-21
> **Response to Reviewer 949b [1/2]**
>
> Thank you so much for your encouraging and highly positive review. We were thrilled to read your feedback and are very grateful for the rating. We truly appreciate that you found our paper well-written and well-motivated, and our experimental results promising. Your thoughtful comments and excellent suggestions have been incredibly helpful, and we’ve enjoyed strengthening the paper based on your advice.
>
> We have carefully revised the manuscript to address all the points you raised. Here are our point-by-point responses:
>
> **Regarding Weakness 1 and Question 5: Direct comparison with other RL methods.**
>
> This is a fantastic suggestion. We completely agree that a direct comparison with other existing RL-based methods is specially important and relevant to explaining our reward design. Following your advice, we have conducted a new ablation study to see how our Sim-RL performs against several other reward methods.
>
> In addition to the simple Binary Reward baseline [1], we also added comparisons with **ToolRL** [2] and **SwiRL** [3]. For SwiRL, we used Qwen3-8B as the Process Reward Model (PRM) due to cost constraints. To keep things fair and really focus on the reward function's impact, all these experiments were run on the same Qwen3-0.6B model that was initialized with our CKD method.
>
> The reward design ablation results, which we've added as **Table 5** in the paper, are shown below:
>
> | Method                | BFCLv3 Overall | ACEBench Normal |
> | --------------------- | -------------- | --------------- |
> | CKD+Binary Reward     | 51.05          | 35.70           |
> | CKD+ToolRL            | 48.59          | 40.50           |
> | CKD+SwiRL             | 51.10          | 40.30           |
> | **CKD+Sim-RL (Ours)** | **51.70**      | **53.00**       |
>
> The results clearly demonstrate that Sim-RL provides a significant advantage over the other methods, especially on the more challenging ACEBench generalization benchmark. Our analysis is as follows:
>
> - **Binary Reward** is too strict. It only rewards rollouts that perfectly match the ground truth, leading to high precision but low recall. This approach struggles in multi-solution scenarios and fails to generalize well, as seen in its poor ACEBench score.
> - **ToolRL** and **Sim-RL** both use a dense reward, but our **Sim-RL** provides a more comprehensive signal by also rewarding non-call text based on similarity, which enhances its generalization capabilities.
> - **SwiRL (The PRM-based approach)** with a Qwen3-8B PRM underperforms Sim-RL. While a more powerful PRM could potentially yield better results, using PRMs introduces significant training costs and the risk of "reward hacking" from a black-box model [4]. Sim-RL, in contrast, is low-cost, interpretable, and empirically more effective in our setup.
>
> This new analysis and the corresponding results have been integrated into **Section 4.3** to better showcase the effectiveness of Sim-RL.
>
>
> **References:**
>
> [1] Hao et al. Exploring Superior Function Calls via Reinforcement Learning, 2025. URL https://arxiv.org/abs/2508.05118v3.
>
> [2] Qian et al. Toolrl: Reward is all tool learning needs, 2025. URL https://arxiv.org/abs/2504.13958.
>
> [3] Goldie et al. Synthetic data generation and multi-step reinforcement learning for reasoning and tool use. In Second Conference on Language Modeling, 2025. URL https://openreview.net/forum?id=oN9STRYQVa.
>
> [4] Khalaf et al. Inference-Time Reward Hacking in Large Language Models, 2025. URL https://arxiv.org/abs/2506.19248.

---

> ### Author Response · Authors · 2025-11-21
> **Response to Reviewer 949b [2/2]**
>
> **Regarding Questions 1-4:**
>
> **1. [Line 106] 057: typo: the -> the:** Thank you for spotting this. We've fixed it in the revised manuscript.
>
> **2. Minor suggestion for structuring Sec 3:** Thanks for your suggestions of improving the paper's flow. In our revised manuscript, we have reordered **Section 3** to introduce CKD (now Section 3.1) before Sim-RL (now Section 3.2), which aligns with the chronological flow of our framework.
>
>
> **3. [Line 106] Add citation for RLVR:** Thanks for the reminder. We have added the appropriate citation.
>
> **4. [Line 270] Clarification on the two iterations of Sim-RL:** Excellent question—this was a point that definitely needed more clarity. You're right, we do use a round of Sim-RL to refine the teacher model. Here’s a bit more detail:
>
> 1. **Why did we do it? Optional, Not Required**: Our goal was to distill from the good teacher possible. By refining the base Qwen3-8B with Sim-RL, we could adapt it to our specific task and create a stronger teacher. This step is optional, not a mandatory part of the framework, but it helps get better performance, as mentioned in Section 3.3. Given the open-source models we had access to, the limited quality of the available data, and the various methods at our disposal (like standard RL, Sim-RL, SFT), we found that the Qwen3-8B model after being refined with Sim-RL was the relatively good teacher we could produce. Therefore, we chose to use this refined teacher to carry out the subsequent work, aiming to transfer the better possible capabilities to our student model.
> 2. **Does it affect our main point? Ablation on Teacher Refinement**: To check this, we ran an ablation study comparing Qwen3-0.6B students distilled from the base teacher vs. the refined teacher, which is shown below. The results show that while a better teacher indeed leads to a better student, this does not affect the overall validity of our method. With the un-refined teacher, our core STAR method (CKD + Sim-RL) still clearly outperformed the standard SFT+Sim-RL baseline. Furthermore, without refinement, the base teacher model is suboptimal. Consequently, the student model's performance after only the CKD stage is comparatively lower. However, the subsequent Sim-RL to this student model results in a substantial performance gain. These observations are crucial as they substantiates the robustness of our framework, demonstrating its effectiveness even when initialized with a less capable teacher model.
>
> Ablation study on teacher model refinement is shown below. We are ready to add this study into the appendix in the future to make this analysis more clear.
>
> | Method                             | BFCLv3 Overall |       | ACEBench Normal |       |
> | ---------------------------------- | -------------- | ----- | --------------- | ----- |
> |                                    | w/o RL         | w/ RL | w/o RL          | w/ RL |
> | SFT+Sim-RL                         | 47.59          | 50.41 | 28.70           | 38.90 |
> | CKD from Qwen3-8B (Base)+Sim-RL    | 47.13          | 51.35 | 31.40           | 47.20 |
> | CKD from Qwen3-8B (Refined)+Sim-RL | 49.56          | 51.70 | 39.00           | 53.00 |
>
> ------
>
> We hope our responses and revisions have clearly addressed all your comments. We really enjoyed incorporating your feedback and feel it has made the paper significantly stronger.
>
> Thank you once again for your support. If you feel these revisions further improve the paper's contribution, we would be absolutely thrilled. We look forward to any further constructive feedback and positive assessment.

---

> ### Comment · Reviewer_949b · 2025-11-24
> **Response to rebuttal**
>
> Thanks for the additional ablation and updating the paper. This will definitely strengthen the results.
>
> Three questions regarding these results:
>
> 1- Are the results on BFCL significantly different or are they within noise?
>
> 2- Do you have qualitative examples or any analysis showing Binary reward leading to higher precision but lower recall in comparison to other method?
>
> 3- Similarly, how did the author investigate reward hacking by SwiRL baseline?

---

> > ### Author Response · Authors · 2025-11-25
> > **Response to Reviewer 949b [1/2]**
> >
> > Thank you for your continued engagement with our work. We appreciate the opportunity to provide clarification on the significance of our results and our analysis of the different reward mechanisms to further strengthen our paper's contributions:
> >
> > >*"1. Are the results on BFCL significantly different or are they within noise?"*
> >
> > This is an excellent question. To ensure our results are robust and not due to random fluctuations, we conducted 3 evaluation runs for each reward method on the BFCLv3 benchmark. The table below shows the mean and standard deviation of the results (the ± value).
> >
> > | Method            | BFCLv3 Overall   | Non-Live Acc     | Live Acc         | Multi Turn Acc  | Irrelevance Detection |
> > | ----------------- | ---------------- | ---------------- | ---------------- | --------------- | --------------------- |
> > | CKD+Binary Reward | 50.83 ± 0.09     | 76.43 ± 0.45     | **69.30 ± 0.15** | 4.50 ± 0.10     | 86.92 ± 0.22          |
> > | CKD+ToolRL        | 49.01 ± 0.13     | 78.43 ± 0.31     | 64.15 ± 0.17     | 4.00 ± 0.37     | 77.00 ± 0.35          |
> > | CKD+SwiRL         | 50.96 ± 0.09     | 77.91 ± 0.10     | 69.18 ± 0.20     | 3.37 ± 0.27     | **87.99 ± 0.21**      |
> > | CKD+Sim-RL (Ours) | **51.80 ± 0.12** | **78.98 ± 0.29** | 68.41 ± 0.30     | **6.93 ± 0.48** | 83.27 ± 0.28          |
> >
> > **Analysis:**
> >
> > The results show that Sim-RL's performance gain is statistically significant and not due to random noise, as confirmed by the non-overlapping score ranges. Binary Reward and SwiRL show a clear advantage in irrelevance handling, like correctly avoiding function calls when unnecessary (boosting their Live Acc.), while ToolRL surpasses them in core function-calling proficiency (leading to a higher Non-Live Acc.). Sim-RL effectively combines both strengths and also establishes a more significant and distinct advantage in multi-turn capabilities, demonstrating its superior robustness in complex interactions.

---

> ### Author Response · Authors · 2025-11-25
> **Response to Reviewer 949b [2/2]**
>
> > *"2. Do you have qualitative examples or any analysis showing Binary reward leading to higher precision but lower recall in comparison to other methods?"*
>
> Yes, this is a key insight into our reward design. A binary reward is overly strict, enforcing an exact match to the ground truth. This leads to a "high precision, low recall" scenario where it only rewards perfect outputs but fails to provide a learning signal for functionally correct or nearly-correct variations. Sim-RL is designed to address this by providing a graded reward.
>
> The table below shows two clear examples of this problem:
>
> |                     | **Example 1: Missing a Default Argument**                    | **Example 2: Trivial Formatting Difference**                 |
> | ------------------- | ------------------------------------------------------------ | ------------------------------------------------------------ |
> | **Function**        | `wordpress.check_wordpress`                                  | `label_template_brands`                                      |
> | **Query**           | "Can you check if `https://example.com` is running WordPress?" | "Can you list the brands available for A4 size blank label sheets?" |
> | **Model Rollout**   | `{"name": "wordpress.check_wordpress", "arguments": {"url": "https://example.com"}} ` | `{"name": "label_template_brands", "arguments": {"format": "a4"}} ` |
> | **Ground Truth**    | `{"name": "wordpress.check_wordpress", "arguments": {"url": "https://example.com", "user_agent": "Mozilla/5.0"}} ` | `{"name": "label_template_brands", "arguments": {"format": "A4"}} ` |
> | **Binary RL Score** | **0** (Mismatch)                                             | **0** (Mismatch)                                             |
> | **Sim-RL Score**    | **0.5** (Partial credit for correct function and primary argument) | **1.0** (ROUGE-L is case-insensitive)                        |
>
> **Analysis:**
>
> The model correctly identified the primary intent and generated a functional tool call. As the table shows, Binary reward's score of 0 is an excessively harsh penalty that provides no useful signal, while Sim-RL correctly identifies that the call is partially correct and provides a granular signal to guide the model toward including the missing optional parameter.
>
>
>
> > *"3. Similarly, how did the author investigate reward hacking by the SwiRL baseline?*"
>
> We investigated reward hacking by the SwiRL baseline by manually analyzing cases where Sim-RL and SwiRL's reward model disagreed. This revealed instances where the SwiRL could be "hacked"—that is, exploited to obtain a high reward for suboptimal or incorrect behavior.
>
> The table below shows a clear example of this:
>
> |                   | **Example: Redundant Tool Call (Reward Hacking)**            |
> | ----------------- | ------------------------------------------------------------ |
> | **Context**       | In a previous turn, the model already looked up information for "SFO" airport. |
> | **Query**         | "What is the ICAO code for SFO airport, and how many runways does it have?" |
> | **Model Rollout** | `<tool_call> {"name": "airportstatistics", "arguments": {"iata": "SFO"}} </tool_call> ` |
> | **Ground Truth**  | `"The ICAO code for SFO is KSFO, and it has 4 runways."`     |
> | **SwiRL Score**   | **1.0** (Rewards the valid-looking tool call, ignoring context) |
> | **Sim-RL Score**  | **0.0** (Penalizes the unnecessary call compared to the optimal response) |
>
> **Analysis:**
>
> This is a classic case of reward hacking. The SwiRL model assigns a high score because the tool call is syntactically correct to the user's immediate question. However, it ignores the conversation's history—the model already has the answer, rewarding an inefficient and redundant action. The model learns to game the system by making an easy, unnecessary tool call. Our Sim-RL compares the action to the optimal ground-truth response and correctly assigns a reward of 0, penalizing this suboptimal behavior.
>
> We hope these detailed responses and examples thoroughly address your questions. We are grateful for your insightful feedback, which has helped us further strengthen the analysis in our paper.

---

### Official Review · Reviewer_ovcd · 2025-11-04

**Soundness:** 3
**Presentation:** 3
**Contribution:** 2
**Rating:** 6
**Confidence:** 3

**Summary:**

This paper proposes a two-stage training framework—a knowledge distillation phase followed by a reinforcement learning (RL) phase—to enable a compact 0.6B Qwen model to achieve strong tool-calling capabilities. For each phase, the authors introduce targeted improvements. In the distillation phase, they use a forward KL divergence variant that augments the standard top-k forward KL to suppress confidently incorrect predictions. In the RL phase, they design a heuristic reward function that evaluates the similarity between rollouts and ground truth with more fine-grained criteria, providing richer reward feedback compared to conventional binary rewards. By combining these techniques, the authors demonstrate that their method effectively trains a small model with strong tool-calling performance.

**Strengths:**

* The paper provides targeted and practical improvements for enhancing tool-calling capability, offering insights that could be useful for others working in this area. The performance gains are solid and well-demonstrated through experiments

**Weaknesses:**

* The framework is sound and the empirical results are solid; however, the methodological contribution is not particularly significant.

**Questions:**

NA

---

> ### Author Response · Authors · 2025-11-21
> **Response to Reviewer ovcd**
>
> We sincerely thank you for your thoughtful and constructive feedback. We are grateful for your summary and recognition of our work's strengths, particularly the practical improvements and solid empirical results. We appreciate your comment regarding the methodological contribution, which highlighted an opportunity for us to provide a more rigorous presentation of our framework's novelty and impact. We have conducted several new experiments and integrated a deeper, more compelling analysis into the revised manuscript, primarily in Section 4.3, with the addition of Figures 3-4 and Table 5, to strengthen the paper and more clearly articulate our contributions.
>
> Our central contribution is the **STAR framework**, a holistic and principled solution for distilling advanced tool-calling capabilities into super-tiny models—a domain where the conventional SFT+RL paradigm often fails due to overfitting and instability. STAR's novelty is rooted in the synergistic integration of two core innovations:
>
> **1. Constrained Knowledge Distillation (CKD) as a Fundamentally Superior RL Initializer:** Our expanded analysis illustrates the superiority of CKD compared to other methods, its synergy with the subsequent Sim-RL phase, and its overall contribution to our STAR framework. CKD is a principled and more effective initializer for the subsequent RL phase. As detailed in the revised **Section 4.3** and supported by our new visualizations (**Figures 3 and 4**), we provide mechanistic and statistical evidence for CKD’s superiority:
>
> - **Superior Reasoning Potential (Higher Pass@k):** CKD endows the initial policy with a significantly higher `Pass@k` accuracy compared to SFT and other KD variants. As established in recent literature [1], this metric is a more robust indicator of a model's true reasoning potential than `Pass@1`, highlighting its capacity to generate a diverse set of correct solutions. This sets a much higher ceiling for the model's final performance.
> - **Enhanced Exploration Capacity (Higher Entropy):** CKD preserves higher policy entropy at the start of RL training. This is a critical advantage, as higher entropy facilitates more effective exploration and prevents premature convergence to suboptimal policies [2]. This is achieved by our unique design, which suppresses "confident-but-wrong" predictions without aggressively collapsing the entire tail of the probability distribution.
>
> **2. Validated Effectiveness of the Sim-RL Reward Function through new ablation studies:** To directly address the significance of our reward design, we have conducted a new ablation study comparing Sim-RL against several strong, established reward functions. As shown in Table 5 in the revised paper, our low-cost, interpretable Sim-RL mechanism delivers a significant performance advantage over **Binary Reward** [3], the reward from **ToolRL** [4], and a PRM-based approach, **SwiRL** [5].
>
> The results of Table 5 in the revised paper demonstrate that Sim-RL provides a richer, more robust, and more effective training signal than prior approaches, especially on the challenging ACEBench generalization benchmark. This is a key contribution for achieving optimal policy refinement in complex, multi-solution tasks like function calling.
>
> In conclusion, the STAR framework's contribution is its principled and synergistic design. CKD creates a superior policy initialization with high reasoning potential and exploratory capacity, and Sim-RL provides the validated, fine-grained reward signal needed to fully realize that potential. Together, they form a powerful and effective blueprint for specializing in super-tiny models.
>
> We believe these new empirical results and deeper analyses provide compelling evidence and present more complete and impactful methodological contributions. We appreciate the opportunity to clarify and strengthen our work based on your insightful feedback. We welcome any further constructive feedback and positive assessment.
>
> Thank you once again for your valuable time and consideration.
>
>
>
>
>
> **References:**
>
> [1] Kang et al. Quagmires in sft-rl post-training: When high sft scores mislead and what to use instead, 2025. URL https://arxiv.org/abs/2510.01624.
>
> [2] Cui et al. The entropy mechanism of reinforcement learning for reasoning language models, 2025b. URL https://arxiv.org/abs/2505.22617.
>
> [3] Hao et al. Exploring Superior Function Calls via Reinforcement Learning, 2025. URL https://arxiv.org/abs/2508.05118v3.
>
> [4] Qian et al. Toolrl: Reward is all tool learning needs, 2025. URL https://arxiv.org/abs/2504.13958.
>
> [5] Goldie et al. Synthetic data generation and multi-step reinforcement learning for reasoning and tool use. In Second Conference on Language Modeling, 2025. URL https://openreview.net/forum?id=oN9STRYQVa.

---

### Author Response · Authors · 2025-11-21
**General response to all reviewers**

Dear Reviewers,

We sincerely thank you for your detailed, insightful, and constructive feedback. We have now uploaded a revised version of our paper, with the changes highlighted for your convenience. While we have provided responses to each reviewer individually, we wish to highlight the key improvements and additions made in this revision:

- **Deeper Analysis of Constrained Knowledge Distillation (CKD) (Section 4.3, Figure 3, Figure 4):** To better elucidate the unique contributions of CKD, we have significantly expanded our analysis in Section 4.3. We now provide a detailed discussion on CKD's advantages over other distillation strategies, its crucial role in preparing a robust policy for the subsequent Sim-RL phase, and its overall significance within the STAR framework. This analysis is supported by new experimental results, including `Pass@k` scores and policy entropy comparisons, which quantitatively demonstrate how CKD preserves vital exploratory capacity for effective RL refinement.
- **Ablation Study on Sim-RL Reward Design (Section 4.3, Table 5):** To rigorously validate our novel reward mechanism, we have introduced a new ablation study comparing our Sim-RL reward design against alternative approaches, such as binary rewards and other specialized methods. The results, presented in the new Table 5, and the accompanying analysis in Section 4.3, empirically confirm that our fine-grained, similarity-based reward provides a more effective and robust signal for policy optimization in complex, multi-solution tasks.
- **Methodological Clarifications and Reorganization (Section 3, Abstract, Introduction):** For improved clarity and logical flow, we have refined the presentation of our methodology. In our revised manuscript, we have reordered the chronological flow of our framework to introduce Constrained Knowledge Distillation (CKD) before Similarity-guided Reinforcement Learning (Sim-RL). We have also clarified in Section 3.2.1 that the use of the Qwen template is an implementation example, underscoring the general applicability of our formatting principles.
- **Enhanced Clarity on Evaluation (Section 4.1, Appendix A.9):** To improve the clarity and reproducibility of our evaluation, we have added a reference in Section 4.1 directing readers to a detailed breakdown of all evaluation categories for the BFCL and ACEBench benchmarks in Appendix A.9.

Thank you again for your valuable input. We believe these revisions have substantially strengthened the paper and addressed the main concerns. We are available for any further questions or discussion.

---

### Author Response · Authors · 2025-12-01
**Rebuttal Summary by the Authors**

Dear Program Chairs, Senior Area Chairs, Area Chairs, and Reviewers,

Thank you for the opportunity to provide a summary of our work and the rebuttal process. Our paper introduces the STAR framework, **a holistic training pipeline designed to distill the advanced function calling capabilities of large language models (LLMs) into super-tiny models (e.g., 0.6B).** This is a significant challenge, as conventional fine-tuning paradigms like SFT+RL often fail for these small models due to overfitting and instability. STAR overcomes this through two synergistic innovations: Constrained Knowledge Distillation (CKD), which creates a high-potential and exploration-ready policy initialization, and Similarity-guided Reinforcement Learning (Sim-RL), which provides a fine-grained reward signal for effective policy refinement. Our work is important as **it not only invites a reconsideration of the standard training paradigm, but also provides a practical blueprint and a superior exemplar for generating optimal super-tiny models using the KD+RL paradigm.** Our STAR-0.6B model finally sets a new SOTA in its size class, even outperforming several much larger models.

We received a diverse set of reviews, and we are grateful for the feedback which pushed us to significantly strengthen our paper. We engaged in a thorough rebuttal process, conducting substantial targeted experiments and deeper analyses to **address every concern** raised by the reviewers. The revised manuscript now contains **new figures (Figures 3-4), new tables (Table 5, Appendix Tables 7-15), and expanded analysis sections (especially Section 4.3),** which we believe provide compelling new evidence for our claims and clarify our contributions.

- **Deeper Mechanistic Analysis and Statistical Validation of CKD.** Regarding the contribution of our distillation method, **Reviewer Qjs5** initially questioned CKD's statistical significance over baselines before RL. To address this, we added a deeper analysis (Section 4.3, Figures 3-4) to provide mechanistic and statistical evidence, arguing that `Pass@1` is an unreliable metric for an RL initializer. Instead, we show **CKD’s superiority in `Pass@k` (reasoning potential) and policy entropy (exploration capacity).** Crucially, in response to **Reviewer Qjs5's** follow-up concerns, we conducted new experiments over 3 random seeds (Appendix A.10), demonstrating with statistical significance that CKD enables far greater performance gains from RL (+12.4 points on ACEBench) compared to strong baselines like AKL (+5.53 points). This robustly validates CKD’s critical role as a superior policy initializer for RL.

- **Rigorous Validation of the Sim-RL Reward Function.** Regarding the effectiveness of our reward design, **Reviewers 949b, Qjs5** rightly requested a direct comparison with other reward functions. We conducted **a comprehensive new ablation study (Table 5), comparing our Sim-RL against established methods** like Binary Reward, ToolRL, and a PRM-based approach (SwiRL). The results clearly show that our low-cost, interpretable Sim-RL provides a significant performance advantage. To further address **Reviewer 949b's** follow-up questions, we provided detailed statistical analysis and qualitative case studies (Appendix A.10-11) demonstrating Sim-RL's superiority in handling partially correct outputs and avoiding reward hacking.

- **Comprehensive Ablations to Verify Framework Design Choices.** Regarding the overall framework design, we addressed several other key concerns to demonstrate the robustness and soundness of our approach. For **Reviewers 949b, u15k**, we clarified that the teacher refinement step is optional and provided an ablation study (Table 15) showing our framework’s effectiveness even with a base, un-refined teacher. For **Reviewer u15k**, we also ran experiments on a larger 1.7B student model (Table 13) and with a larger 14B teacher (Table 14) to demonstrate scalability. For **Reviewer Qjs5**, we provided a new experiment showing that distilling on ground-truth data (without teacher-generated reasoning) leads to poor performance, validating our data strategy and we also tested an alternative symmetric divergence (JSD), showing CKD is still superior for our task. These extensive new studies validate the key design principles of the STAR framework.

- **Addressing Other Concerns.** We also updated our manuscript, incorporating all minor suggestions regarding presentation, citations and the chronological flow of CKD and Sim-RL.

**Through targeted experiments and deeper analysis, we have addressed all major reviewer concerns.** The revised manuscript now presents a well-validated, robust, and significant contribution. We are confident that STAR provides a powerful and principled blueprint for specializing super-tiny models—an area of great practical importance—and we hope you will find the improved paper worthy of acceptance. Thank you for your time and consideration.

Best regards,

The Authors

---

### Meta-Review · Area_Chair_HdXR · 2026-01-07

**Summary:**

The paper proposes STAR, a two-stage KD+RL pipeline for super-tiny function-calling models, with two main components: Constrained Knowledge Distillation (CKD) to produce a stable, exploration-capable initializer, and Similarity-guided RL (Sim-RL) to provide a fine-grained reward for multi-solution tool-calling tasks. Reviewers’ key questions centered on (i) whether CKD’s benefits are robust and meaningful beyond standard KD/SFT initializers, and (ii) whether Sim-RL is clearly differentiated from and superior to prior on-policy RL reward designs.

In the rebuttal/revision, the authors add direct comparisons against other on-policy reward methods (Binary Reward, ToolRL reward, and a PRM-based SwiRL variant) under a controlled CKD-initialized setting, where Sim-RL shows a substantial advantage—most notably on ACEBench generalization. Along with added analyses on why CKD is a stronger RL initializer and improved ablations/clarifications, the updated evidence supports the central claims, and I recommend acceptance.

**Reviewer Concerns:**

Addressed by the rebuttal / revision:
- Lack of direct comparison to existing RL-based reward methods: addressed via a new ablation comparing CKD+Sim-RL against CKD+Binary Reward, CKD+ToolRL, and CKD+SwiRL, showing Sim-RL’s strongest performance, particularly on ACEBench generalization.
- Clarifications on training curriculum / chronology (KD then RL) and the teacher-refinement step: authors clarify the pipeline structure and note teacher refinement is optional, with supporting ablations.
- Requests for stronger evidence around CKD’s role as an RL initializer: authors expand analysis (e.g., Pass@k/entropy discussion and multi-seed/statistical validation) to argue CKD better preserves exploration and yields larger downstream RL gains.

Still outstanding / partially addressed:
- Broader external validation: while the new controlled ablations strengthen the claim that Sim-RL is better than several strong reward baselines, evidence remains primarily within the chosen model family/datasets; additional independent settings (different student architectures or broader tool suites) would further strengthen generality.
- Methodological novelty perception:  the overall conceptual leap still feels incremental.

**Reviewer Scores:**

Reviewer ovcd: 6 -> 6, Reviewer 949b: 8 -> 8, Reviewer u15k: 6 -> 6, Reviewer Qjs5: 2 -> 5.

---

### Decision · Program_Chairs · 2026-01-26

Accept (Poster)